# Structural analysis of the overoxidized Cu/Zn-superoxide dismutase in ROS-induced ALS filament formation

Yeongjin Baek[1], Tae-Gyun Woo[2], Jinsook Ahn ◉ [1,3,4], Dukwon Lee[1], Yonghoon Kwon ◉ [1], Bum-Joon Park ◉ [2] & Nam-Chul Ha ◉ [1,3 ✉]

Eukaryotic Cu, Zn-superoxide dismutase (SOD1) is primarily responsible for cytotoxic filament formation in amyotrophic lateral sclerosis (ALS) neurons. Two cysteine residues in SOD1 form an intramolecular disulfide bond. This study aims to explore the molecular mechanism of SOD1 filament formation by cysteine overoxidation in sporadic ALS (sALS). In this study, we determined the crystal structure of the double mutant (C57D/C146D) SOD1 that mimics the overoxidation of the disulfide-forming cysteine residues. The structure revealed the open and relaxed conformation of loop IV containing the mutated Asp57. The double mutant SOD1 produced more contagious filaments than wild-type protein, promoting filament formation of the wild-type SOD1 proteins. Importantly, we further found that HOCl treatment to the wild-type SOD1 proteins facilitated their filament formation. We propose a feasible mechanism for SOD1 filament formation in ALS from the wild-type SOD1, suggesting that overoxidized SOD1 is a triggering factor of sALS. Our findings extend our understanding of other neurodegenerative disorders associated with ROS stresses at the molecular level.

[1] Department of Agricultural Biotechnology, and Research Institute of Agriculture and Life Sciences, CALS, Seoul National University, Seoul 08826, Republic of Korea. [2] Department of Molecular Biology, College of Natural Science, Pusan National University, Busan 46241, Republic of Korea. [3] Center for Food and Bioconvergence, Seoul National University, Seoul 08826, Republic of Korea. [4] Present address: Center for Biomolecular and Cellular Structure, Institute for Basic Science (IBS), Daejeon 34126, Republic of Korea. ✉email: hanc210@snu.ac.kr

Many neurodegenerative disorders, such as Parkinson's and Alzheimer's disease and amyotrophic lateral sclerosis (ALS), are associated with protein aggregates as a common pathological feature[1]. ALS leads to adult-onset degeneration of motor neurons in the spinal cord and cortex. ALS is mainly described as a sporadic case with no genetic mutation (sporadic ALS, sALS), and only ~10% of cases are genetically linked (familial ALS, fALS). More than 150 mutations at the superoxide dismutase 1 (SOD1) gene were discovered in fALS instances and were associated with 20% of fALS cases[2]. For instance, the G93A mutation at the SOD1 gene resulted in the formation of cytotoxic filamentous forms from the soluble pool in neuron cells[3–5]. Transgenic mice harboring the human G93A mutant SOD1 gene showed typical ALS symptoms with the structural transition of the soluble SOD1 protein to the filamentous forms[6,7].

The wild-type SOD1 protein is converted to the cytotoxic filament under specific but not fully understood environmental stresses in sALS cases[8–10]. One of the leading causes of sALS is oxidative stress, including aberrant free radical metabolism[11–15]. Notably, extreme physical exercises are a well-known risk factor for sALS[16]. Previous studies have suggested an increased risk of ALS among athletes and people who engage in intense physical activity[17–25]. To date, the stimuli that trigger filament formation of the wild-type SOD1 proteins at the molecular level remain to be elucidated.

Eukaryotic SOD1 is a 32-kDa homodimeric metalloenzyme containing Cu and Zn ions at the active site in the active state. The protomer of SOD1 forms an eight-stranded Greek-key β-barrel fold with four conserved cysteine residues and three long loops (loops IV, VI, and VII). The Cu and Zn ions are bound at the active site by loop IV, whose conformation is stabilized by the highly conserved intramolecular disulfide bond between Cys57 in loop IV and Cys146 in 8th β-strand (β8). The cleavage of the intramolecular disulfide bond and the loss of metal ions resulted in the metal-free denatured monomer, which is the entry point to aggregation for filament formation in pathological states[26]. Without a reducing agent, the SOD1 protein rarely forms oligomers in vitro.

The human copper chaperone for SOD1 (hCCS) forms a transient complex with monomeric SOD1, catalyzing copper binding to SOD1 and mediating disulfide bond formation between Cys57 and Cys146 within the subunit, leading to the formation of dimeric holo-SOD1. This hCCS-dependent disulfide bond formation of SOD1 requires molecular oxygen; thus, hypoxic conditions hinder hCCS functions[27,28]. A recent study showed that hypoxic stress on the cells induced a disulfide reduction, thereby increasing the structural disorders of SOD1[28]. Furthermore, other studies have shown that the disulfide-forming cysteine in SOD1, especially Cys146, is prone to be overoxidized. A previous study has shown that the Cys146 that is mutated in familial ALS[29], is oxidized to cysteine-sulfonic acid in AD and PD brains[30]. In addition, a high-resolution mass spectrometric analysis also revealed that yeast Sod1 is oxidized at Cys146 and His71 upon sustained expression under oxidative conditions[31].

Cysteine is oxidized into three different structures depending on the oxidation states: cysteine-sulfenic acid, cysteine-sulfinic acid, and cysteine-sulfonic acid (Fig. 1a and supplementary Fig. 1). Cysteine-sulfenic acid can be reduced back to cysteine in the presence of reducing agent or makes a disulfide with cysteine. Cysteine-sulfinic acid can be reduced only by special enzymes, such as sulfiredoxin, requiring ATP hydrolysis[32]. It has not been found that cysteine-sulfonic acid cannot be reduced under the normal conditions[33,34]. Aspartic acid mimics cysteine-sulfinic acid in terms of the electrostatic charge and the atomic arrangement. Both have one negative charge and two oxygen atoms attached to the central atom (C or S) in the triangular plane geometry. We previously confirmed that the mutation of Cys to Asp well mimics the cysteine-sulfinic acid or cysteine-sulfonic acid in the high-resolution crystal structures of the bacterial redox sensor OxyR[35].

To investigate the molecular mechanism for sALS instances with the wild-type SOD1 gene, we focused on overoxidation at Cys57 and Cys146 in facilitating the nucleation and growth of the filament formation of SOD1. The crystal structure and biochemical studies of the overoxidation-mimicking mutant (C57D/C146D) SOD1 proposed a new mechanism linking physiological stimuli to filament formation at the molecular level.

## Results

**Monomeric overoxidation-mimicking C57D/C146D mutant SOD1.** Cysteines can be oxidized into disulfide, which is highly resistant to further oxidation in the physiological state. However, cysteines are also vulnerable to overoxidation into the cysteine-sulfinic or cysteine-sulfonic acid forms under certain oxidative environments. To investigate the possible roles of the overoxidation of cysteine residues in SOD1, a mutant SOD1 protein that could mimic the overoxidized Cys57 and Cys146 was generated. The C57D/C146D mutant SOD1 proteins were well expressed in the E. coli expression system as the wild-type SOD1 protein.

To biochemically characterize the overoxidation-mimicking mutant SOD1, we first compared the metal contents of the as-isolated mutant and wild-type SOD1 proteins purified using the same procedure. The ICP-MS analysis revealed that the mutant SOD1 proteins almost lost the bound $Cu^{2+}$ and $Zn^{2+}$ compared to the wild-type SOD1 proteins (Table 1). This result showed that the protein sample contained only a small portion of the holoenzyme containing $Zn^{2+}$ and $Cu^{2+}$ and further indicates that the C57D/C146D mutant SOD1 protein has lower affinities for $Zn^{2+}$ and $Cu^{2+}$ than the wild-type SOD1 protein.

To examine the size and shape of the C57D/C146D mutant SOD1 in the purified protein sample, size-exclusion chromatography combined with multiangle light scattering (SEC-MALS) was carried out. The MALS analysis, which measured the shape-independent molecular size, demonstrated that the C57D/C146D mutant SOD1 protein was mostly monomeric in solution, whereas all the wild-type SOD1 proteins were dimeric (Fig. 1b). However, the mutant protein eluted faster in size-exclusion chromatography in a broad peak than wild-type proteins in a sharp peak, indicating that the mutant protein exhibited a more relaxed and open conformation as a monomer than the compact and dimeric wild-type protein. Interestingly, ~30% of the purified wild-type SOD1 protein was in reduced form with free thiols at Cys57 and Cys146 in the absence of reducing agents in the sample buffer (Supplementary Fig. 2a). The dimeric assembly of disulfide-reduced wild-type SOD1 was also confirmed through the SEC-MALS in the 5 mM DTT-containing buffer (Supplementary Fig. 2b). Thus, our findings showed that both the wild-type SOD1 protein is dimeric regardless of disulfide bond between Cys57 and Cys146, unlike the monomeric C57D/C146D mutant SOD1 protein.

**Less structurally stable C57D/C146D mutant SOD1 protein with a lower activity.** To analyze the structural stability of the C57D/C146D mutant SOD1 protein, the melting temperature (Tm) values of the wild-type and mutant SOD1 proteins were measured by the thermal shift assay. In this experiment, we used the C57A/C146A mutant protein as the disulfide incapable SOD1 protein. Compared to those of the wild-type and C57A/C146A mutant SOD1 proteins, the Tm value of the C57D/C146D mutant

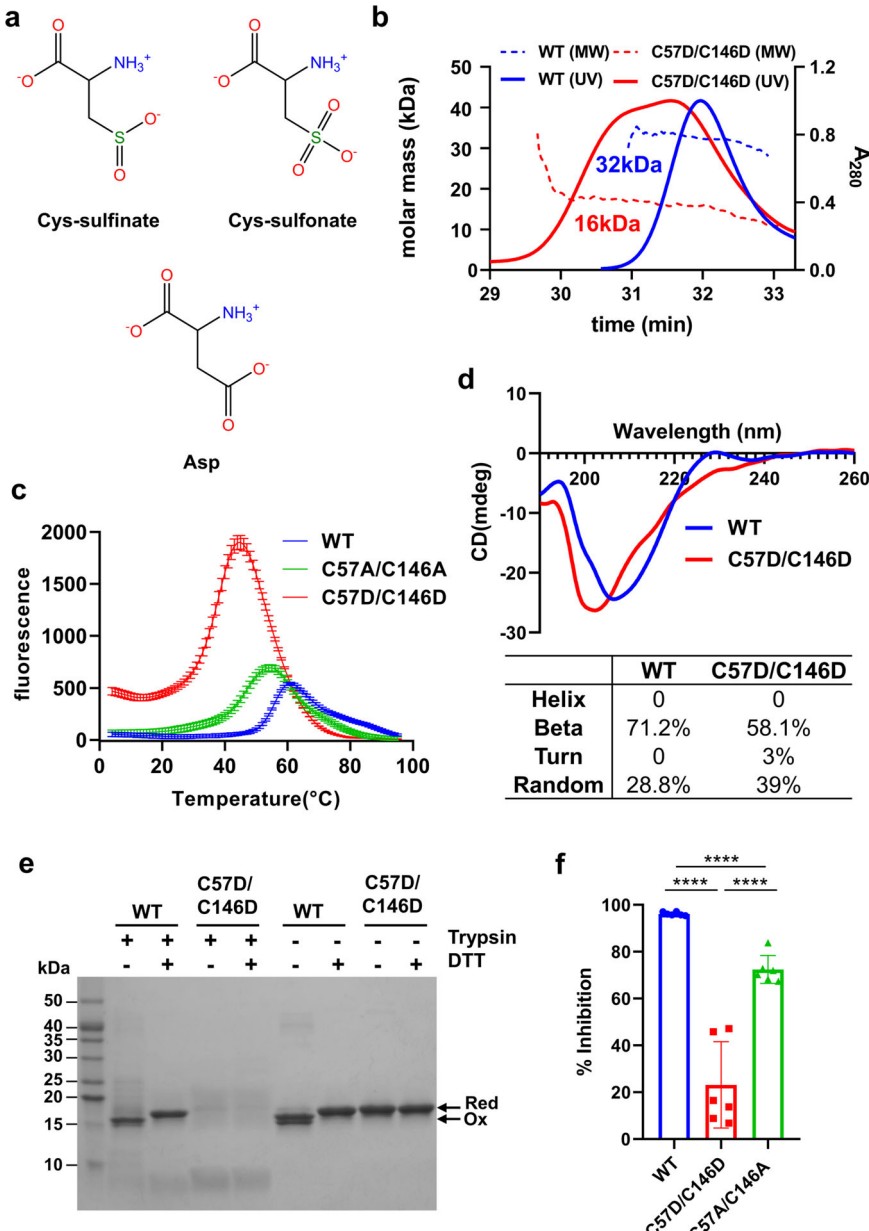

**Fig. 1 Effect of the C57D/C146D mutation on the biochemical properties of purified SOD1 proteins. a** Structural comparison of cysteine-sulfinic acid and cysteine-sulfonic acid to aspartic acid. Cysteine-sulfinic acid and -sulfonic acid are the overoxidized forms of cysteine in the presence of ROS. Note that aspartic acid mimics Cys-sulfinic acid and Cys-sulfonic acid in terms of the electrostatic charge and the atomic arrangement. Detailed descriptions are in Supplementary Fig. 1. **b** SEC-MALS profiles of the purified wild-type (blue) and C57D/C146D mutant (red) SOD1 proteins. The UV absorbance at 280 nm ($A_{280}$ at the right Y-axis) of the SEC is represented by solid lines. The molecular mass (the left Y-axis) based on MALS is represented by a dotted line. The average molar mass of wild-type or C57D/C146D mutant SOD1 is indicated under the dotted lines. **c** Tm measurement by the thermal shift assay of the wild-type SOD1 protein (WT, blue line), C57A/C146A mutant SOD1 protein (green line), and C57D/C146D mutant SOD1 protein (red line). Individual raw data points displayed for each temperature represented the average of three endpoint readings with ±SD. According to the analyzing results (Supplementary Fig. 2c), the Tm values of the wild-type, C57A/C146A, and C57D/C146D were 56 °C, 47 °C, and 38 °C, respectively. **d** *Top*, far-UV CD spectra of the wild-type SOD1 (WT, blue line) and C57D/C146D mutant SOD1 (red line) at the wavelength range of 190-260 nm. *Bottom*, the estimated secondary structure of wild-type and mutant protein calculated based on CD spectra. The proportion of each secondary structure constituting the SOD1 protein is expressed as a percentage. **e** Proteolytic digestion of the wild-type (WT) and C57D/C146D mutant SOD1 proteins by trypsin. The proteins were digested at 37 °C for 1 h in the presence or absence of 5 mM DTT and then subjected to SDS-PAGE. The redox states of SOD1 proteins are indicated by arrows to the right of the SDS polyacrylamide gel image. All the cysteines in SOD1 are reduced (red), or an intramolecular bond is formed between Cys57 and Cys146 (Ox). **f** Relative superoxide dismutase activities of the wild-type (blue), C57D/C146D (red), and C57A/C146A (green) mutant SOD1 protein. Inhibition of cytochrome *c* reduction by superoxide in the presence of SOD1 was represented in bar graphs with mean ± SD from six individual experiments. Statistical comparisons were performed using an unpaired two-tailed Student's *t*-test. Differences were considered statistically significant at *P*-values < 0.05. **** denote *p* < 0.0001. Raw data are presented in Supplementary Fig. 2e.

| Table 1 Copper and zinc content analysis of wild-type and C57D/C146D mutant SOD1 proteins revealed by ICP-MS. | | |
|---|---|---|
| **SOD1** | **Copper** | **Zinc** |
| wild-type | 26.3% | 64.9% |
| C57D/C146D | 1.4% | 6.7% |
| The percentages of the bound metal per SOD1 monomer are shown in a table. | | |

SOD1 protein was decreased by ~20 °C and ~10 °C, respectively (Fig. 1c). The CD spectroscopic results demonstrated that the C57D/C146D mutant SOD1 was more disordered, having a smaller β-sheet portion and a more random coil portion than the wild-type SOD1 (Fig. 1d).

Consistent results were produced with the high sensitivity of the C57D/C146D mutant SOD1 proteins to proteolysis (Fig. 1e). When the wild-type and C57D/C146D mutant SOD1 were treated with trypsin, the C57D/C146D mutant SOD1 was rapidly degraded compared to the wild-type SOD1 proteins both in the absence and presence of the reducing agent. The wild-type SOD1 in the presence of the reducing agent was highly resistant to trypsin digestion regardless of the reducing agent in the buffer, indicating the conformation of the wild-type SOD1 did not depend on the disulfide bond. These results are also supported by the crystal structure of the C57A/C146A mutant SOD1, which could not make the disulfide bond[36]. The C57A/C146A SOD1 mutant protein showed a higher proteolytic resistance than the C57D/C146D mutant SOD1 (Supplementary Fig. 2d). Therefore, overoxidation is required for the structural change of SOD1, together with the break of the disulfide bond between Cys57 and Ctys146.

Next, the enzyme activity of the C57D/C146D mutant SOD1 was compared with C57A/C146A mutant SOD1 or wild-type SOD1 proteins using purified protein samples. The enzymatic activity of the C57D/C146D mutant SOD1 was nearly abolished in the SOD activity assay, which is based on the optical absorbance from the cytochrome $c$ protein reduced by superoxide (Fig. 1f). The enzyme activity of C57D/C146D mutant SOD1 was even lower than the activity of C57A/C146A mutant SOD1, which indicates that the overoxidation of SOD1 further diminish the enzyme activity than the only disulfide breakage. The mutant SOD1-supplemented sample accumulated the superoxide anions generated by an NADH oxidase (a recombinant RclA mutant protein[37]), as much as the SOD1-free control (Fig. 1f and Supplementary Fig. 2e). These results show that the C57D/C146D mutant SOD1 has lower activity than the wild-type or disulfide-cleaved mutant SOD1 (C57A/C146A).

**C57D/C146D mutant SOD1 induces cellular aggregation.** To evaluate the filament-forming ability of the C57D/C146D mutant SOD1 protein in neuronal cells, wild-type or mutant SOD1 was ectopically expressed in the human neuroblastoma cell line SK-N-SH. The ectopic SOD1 proteins in the cell lysate were analyzed to measure the cellular misfolded SOD1 by the dot blotting assay using the antibody specifically recognizing the misfolded SOD1 protein[38,39]. The results from the dot blotting showed that the overexpression of the mutant SOD1 produced misfolded SOD1 proteins more than that of wild-type SOD1 (Fig. 2a). Since SOD1 filaments were shown in the misfolded forms in the cells, our findings indicate that the overoxidation-mimicking C57D/C146D mutant SOD1 exhibits a higher propensity to form filaments in the cellular environment. Furthermore, when cells were further treated with $Cu^{2+}$ or $Zn^{2+}$ chelators to promote filament formation by abstracting metal ions from cellular SOD1, the mutant SOD1 protein also showed a higher propensity to form filaments

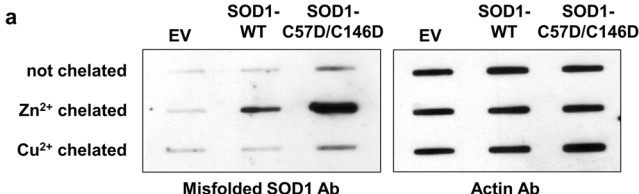

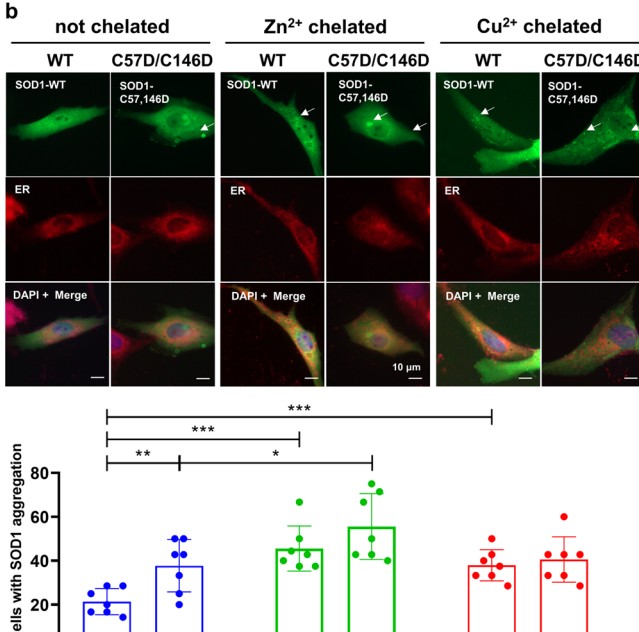

**Fig. 2 Cellular inclusions of the ectopically expressed SOD1 proteins in SK-N-SH cells. a** The dot blot analysis of the misfolded SOD1 ectopically expressed in SK-N-SH cells. To detect the misfolded SOD1 in the cell lysate, the anti-misfolded SOD1 antibody was used (left). SK-N-SH cells were transfected with pcDNA3.1 plasmid (EV), pcDNA3.1 expressing wild-type SOD1 (SOD1-WT), or pcDNA3.1 expressing C57D/C146D mutant SOD1 (SOD1-C57D/C146D). The $Zn^{2+}$ chelator (TPEN) or the $Cu^{2+}$ chelator (ATN-224) was treated in the cells during SOD1 transfection. Actin was detected by the actin antibody as a loading control (right). **b** *Top*, immunofluorescence staining of overexpressed SOD1 in SK-N-SH cells. SK-N-SH cells transfected with pcDNA3.1 expressing the wild-type SOD1 or the C57D/C146D SOD1 gene were incubated with $Cu^{2+}$ chelator (ATN-224, right panels) or $Zn^{2+}$ chelator (TPEN, middle panels), or without any treatment (left panels) for 12 h. Cells were visualized by the appropriate antibody and dyes: SOD1 in green, ER in red, and DNA in blue. Colocalization of SOD1 with ER shows an orange field. Arrows indicate the cytoplasmic accumulation of SOD1 proteins. White arrows indicated cells with SOD1 inclusions. Scale bar; 10 μm. *Bottom*, statistical quantification of the inclusion positive cells detected by immunofluorescence staining. The percentages of inclusion positive cells to the total cells were represented in bar graphs with mean ± SD from seven biological replicates. Statistical comparisons were performed using Student's $t$-test. Statistical comparisons were performed using Student's $t$-test. $P$-value < 0.05 was considered significant. *, **, *** denote $p < 0.05$, $p < 0.01$ and $p < 0.001$, respectively.

than the wild-type SOD1 protein in the dot blotting assay (Fig. 2a).

We next evaluated SOD1 aggregation in neuronal SK-N-SH cells by light microscopy (Fig. 2b and Supplementary Fig. 3). Ectopic expression of the C57D/C146D mutant SOD1 gene induced more cytosolic inclusions in cells without the treatment of the metal chelators than the wild-type SOD1, indicating that

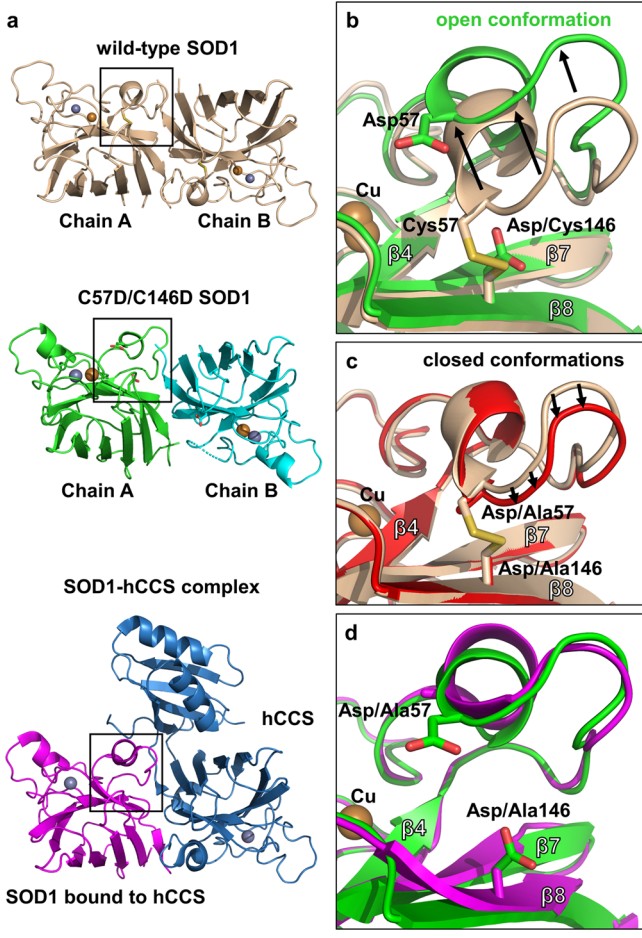

**Fig. 3 Crystal structure of the C57D/C146D mutant SOD1 at 1.8 Å resolution. a** Structural comparison of C57D/C146D mutant SOD1 (chain A;green, chain B; cyan) with wild-type holo-SOD1 dimer (wheat) or with C57A/C146A mutant SOD1 (magenta) bound to human copper chaperone of SOD1 (hCCS, blue) (6FP6). The structure of the asymmetric unit of C57D/C146D mutant SOD1 consists of two protomers (chains A and B). Each subunit is depicted by the ribbon representation. The gold spheres represent $Cu^{2+}$, and the gray spheres represent $Zn^{2+}$. The loop IV regions are marked as three small boxes. **b** Loop IV region in the C57D/C146D mutant SOD1 structure aligned with wild-type SOD1. The C57D/C146D mutant SOD1 is colored green, and the wild-type SOD1 structure is colored wheat. The secondary structural elements are labeled. The gold sphere represents $Cu^{2+}$. Each residue is labeled and indicated in the stick representations with the ribbon diagram of the background. **c** Loop IV region in the C57A/C146A mutant SOD1 structure (6FOI) aligned with wild-type SOD1. The C57A/C146A mutant SOD1 is colored red, and the wild-type SOD1 structure is colored wheat. The secondary structure elements are labeled. The gold sphere represents $Cu^{2+}$. Each residue is labeled and indicated in the stick representations with the ribbon diagram of the background. **d** Loop IV region of the C57D/C146D mutant SOD1 structure aligned with the C57A/C146A mutant SOD1 structure bound to the hCCS (6FP6). The C57D/C146D mutant SOD1 is colored green, and the C57A/C146A mutant SOD1 is colored magenta. The secondary structural elements are labeled. The gold sphere represents $Cu^{2+}$. Each residue is labeled and indicated in the stick representations with the ribbon diagram of the background.

the C57D/C146D mutation contributes to the filament formation of SOD1 in the cell (Fig. 2b, not chelated). The inclusions were further increased in both the wild-type and mutant SOD1-transfected cells when the $Zn^{2+}$ chelator was treated in the cell, indicating the importance of $Zn^{2+}$ in the structural stability

(Fig. 2b, $Zn^{2+}$-treated). However, when the $Cu^{2+}$ chelator was treated, both wild-type and the mutant SOD1 inclusions were as increased as the C57D/C146D mutant SOD1 gene without the chelators, probably because the lower binding affinity of the mutant SOD1 protein to $Cu^{2+}$ (Table 1) (Fig. 2b, $Cu^{2+}$-chelated). Thus, our results suggest that the C57D/C146D mutant SOD1 is more vulnerable to the structural transition to the filamentous form, presumably by overoxidation at Cys57 and Cys146.

**Crystal structures of SOD1 account for the increased monomeric and flexible nature by cysteine-overoxidation.** To structurally relate the cysteine-overoxidation effects, the crystal structures of wild-type SOD1 and overoxidation-mimicking the C57D/C146D mutant SOD1 were compared. The wild-type SOD1 structure was determined at 2.7 Å resolution, which contained twelve molecules and two partial models in the asymmetric unit (Supplementary Fig. 4a). The twelve SOD1 protomers were organized in a honeycomb arrangement with a large space, as previously observed (Supplementary Fig. 4b)[40–43]. The complete structures of wild-type SOD1 were also nearly identical to the SOD1 structures previously determined[42,44–48]. Interestingly, the partial and indeterminate models occupied the large space between the honeycomb arrangement with the highly diffused 2Fo-Fc electron density maps, reflecting the structural flexibility of SOD1 (Supplementary Fig. 4b–d).

We subsequently crystallized the mutant protein sample containing a small portion of the metal-bound SOD1 proteins, as mentioned. The crystal structure of the C57D/C146D mutant SOD1 determined at 1.8 Å resolution showed the metal-bound SOD1 dimeric structure at the active site, which contained one dimer in the asymmetric unit (Fig. 3a and Supplementary Fig. 5a). This result indicated that the metal-bound SOD1 was crystallized despite the small portion of the metal-bound form in the protein sample (see Table 1). The monomeric form of the metal-free monomeric forms does not seem to be crystallized due to the intrinsic structural flexibility.

The metal-containing state of the C57D/C146D mutant SOD1 adopted a similar overall structure to the wild-type SOD1, with an RMSD of 1.311 Å between the protomers (residues 2-153) of 152 Cα atoms (Supplementary Fig. 5b). Loop IV containing the mutated Asp57 residue is not involved in the packing interaction in the crystal, representing the protein conformation in the solution. Notably, the loop was disordered (chain B) or significantly displaced from β8 containing mutated Asp146, resulting in the open conformation at loop IV (chain A; Fig. 3b). The open conformation of loop IV in the mutant SOD1 was distinct from the wild-type and the previous C57A/C146A mutant SOD1 structure (Fig. 3b and c)[36]. The conformation of loop IV in the mutant SOD1 was similar to the open conformation of the SOD1 monomer in a heterodimeric complex with the hCCS monomer, which was induced by the molecular interaction with hCCS (Fig. 3c)[36].

Since the loop IV conformation of the C57A/C146A mutant SOD1 was similar to that of the wild-type SOD1 (Fig. 3c), the built-up negative charges may be responsible for the open conformation of loop IV in the C57D/C146D mutant SOD1 protein. Charge repulsion between Asp57 and Asp146 appeared to induce conformational differences in loop IV in the overoxidation-mimicking mutant. Thus, our findings suggest that the negative charges at the cysteine residues promote the structural changes at loop IV, distinguishing them from the free thiol forms at the Cys57 and Cys146 sites. Since the loop IV is in the dimer interface, the closed conformation loop IV was important for the stabilization of dimeric SOD1.[49–51] The open conformation of loop IV in the C57D/C146D mutant protein

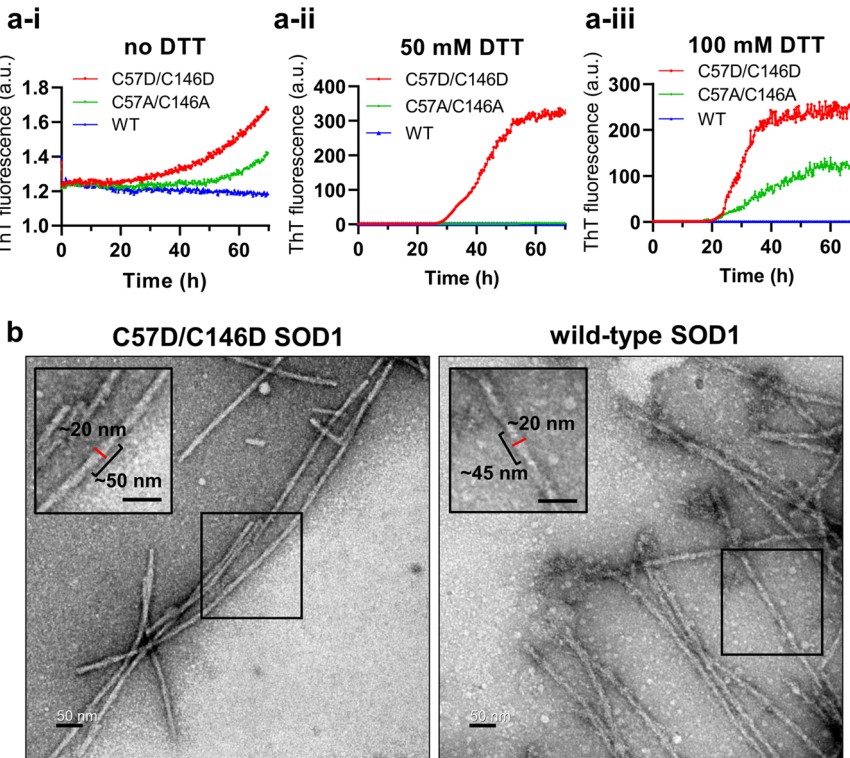

**Fig. 4 Amyloid-like filament formation of the SOD1 proteins. a** Thioflavin T (ThT) fluorescence intensities representing the filament formation of the C57D/C146D mutant SOD1 (red), C57A/C146A mutant SOD1 (green), and wild-type SOD1 (WT, blue) proteins in the reaction buffer containing 0 mM DTT (A-i), 50 mM DTT (A-ii), and 100 mM DTT (A-iii). The data points are represented by the mean values of six independent experiments. The individual raw datasets are presented in Supplementary Fig. 6a. **b** Negative-stain transmission electron micrographs of the C57D/C146D mutant SOD1 (left) or wild-type SOD1 (right) proteins. Both proteins were incubated in reaction buffer containing 50 mM DTT for 90 h. The boxes within the micrographs are enlarged in the corners of the micrographs. Scale bars indicate 50 nm.

indicates the increased monomeric and flexible nature of the cysteine-overoxidation-mimicking mutant SOD1, which is consistent with the MALS results.

The Arg143 residue near loop IV and the bound $Cu^{2+}$ at the active site played an important role in catalysis by offering the microenvironment for superoxide recruitment to the active site[30,52]. The arginine residue is immobilized by the interaction with loop IV in the wild-type SOD1 structure (Supplementary Fig. 5c). In the mutant structure, the arginine residue is disordered by lacking the interaction with disordered or displaced loop IV, as observed in the hCCS-complex structure[36]. This increased conformational flexibility of Arg143 in the C57D/C146D mutant may account for the low catalytic activity of the metal-bound mutant SOD1.

**Overoxidation-mimicking mutation facilitates filament formation.** The role of the structural change of SOD1 by overoxidation was investigated in terms of the filament-forming ability, which was evaluated with purified proteins based on the amyloid-binding dye thioflavin T (ThT) assay. The ThT assay has been used to detect amyloid-like filament formation of SOD1 because fluorescence intensity is increased upon binding of ThT to amyloid-like filaments[53,54].

We first compared filament formation of the C57D/C146D mutant SOD1 protein with that of the wild-type or C57A/C146A mutant SOD1 protein in the absence of the reducing agent (Fig. 4a–i). The C57D/C146D mutant protein formed amyloid-like filaments more strongly than the C57A/C146A mutant SOD1. The negative-staining electron microscopy (EM) image with the C57D/C146D mutant SOD1 showed typical SOD1 amyloid filaments, confirming the formation of amyloid-like

filaments with a good correlation with the results from the ThT assay (Fig. 4b). However, the wild-type SOD1 did not create amyloid-like filaments in the absence and presence of the reducing agent. In an independent experiment with the same kinds of wild-type and C57D/C146D mutant SOD1 proteins, which have longer purification steps, the wild-type SOD1 protein formed filaments in the buffer containing the reducing agent above 50 mM albeit weaker than the mutant proteins (Supplementary Fig. 6b and c). Maintaining a high portion of the reduced form might require forming the filament with the wild-type SOD1 protein, which is different from the C57D/C146D and C57A/C146A mutant SOD1 proteins requiring no reducing agent. Moreover, the rate and intensity of filament formation were generally proportional to the concentration of DTT even in the mutant proteins (Fig. 4a-ii and a-iii).

Our results showed that the cleavage of the disulfide bond of Cys57 and Cys146 by overoxidation, accompanying the structural change in loop IV, is vital in forming the filament. The overoxidation of SOD1 would lower the activation barrier for structural transition to the filamentous forms with charge repulsion between the overoxidized Cys57 and Cys146.

**Filament structures of the wild-type and overoxidation-mimicking mutant SOD1.** To visualize the filaments from the wild-type and mutant SOD1, we employed negative-stain electron microscopy (EM) for the filaments of each protein. The negative-stain EM image showed typical SOD1 amyloid filaments, thus confirming the formation of amyloid-like filaments with a good correlation with the results from the ThT assay (Fig. 4b and Supplementary Fig. 7). The filament of wild-type SOD1 was formed under only experimental conditions containing over

50 mM DTT (Supplementary Fig. 7a and c). The filaments from the C57D/C146D mutant SOD1 showed similar morphology regardless of the presence of DTT (Supplementary Fig. 7b and d). Both filamentous wild-type and mutant SOD1 proteins exhibited similar twisted string features with ~20 nm widths and ~ 45–50 nm helical pitches (Fig. 4b). Our findings suggest that mutant SOD1 shares the filament formation mechanism with the wild-type SOD1 proteins.

**Overoxidized SOD1 as an ALS triggering factor candidate.** Nucleation and growth steps are separately required for ordered oligomer formation[55]. In many cases, the nucleation step determines the rate of oligomer formation in a different process from the growth step. Thus, when a small amount of the pre-existing filament was added, the rate and strength of filament formation were significantly increased, which is called the 'seed effect' in this study. The seed effects of the filaments from the wild-type and C57D/C146D mutant SOD1 proteins were compared. The same amounts of the exponentially growing filament of the wild-type or the mutant SOD1 were taken as the seed (Supplementary Fig. 8a). We then seeded filament formation on the wild-type and mutant SOD1 proteins in the absence and presence of 50 mM DTT with 5% seeds (mass ratio to the soluble protein).

Notably, the C57D/C146D mutant filaments promoted the formation of amyloid-like filaments of the SOD1 wild-type protein regardless of the addition of DTT (Fig. 5a). In contrast, the wild-type SOD1 filament showed a weaker seed effect than the mutant seeds. Thus, our results indicate that the overoxidation-mimicking mutant is a more robust and potent filament-promoting factor than the wild-type filament. Unexpectedly, the wild-type filament helped the formation of the filament on the different proteins (the mutant SOD1 protein) as a nucleation seed better than on the same protein (wild-type SOD1 protein) (Supplementary Fig. 8c). In other words, the wild-type filament exhibited the most negligible seed effect on the wild-type protein, in which both the filament and the soluble protein do not have negative charges on the loop. Therefore, our findings suggest that negative charges are required through electrostatic interactions when the new molecule of reduced SOD1 is recruited at the filament-growing interface.

To gain structural insights into the difference in seed effects, the filament structures from the wild-type and mutant SOD1 proteins were analyzed by the negative-staining EM method (Fig. 5b and Supplementary Fig. 9a–d). As expected, the wild-type-seeded wild-type filament and mutant-seeded mutant filaments showed the same morphology as the unseeded wild-type filament and mutant filaments, respectively (Fig. 5b-i, Supplementary Fig. 9c and d). All the mutant filaments produced by seeded filamentation showed similar morphology, with filaments formed in the absence of seeds (Supplementary Fig. 8a–d). In contrast, only the structures of short-piled filaments were observed in the wild-type filament-seeded wild-type SOD1 protein, which may explain the most negligible seed effect of the wild-type filament on the wild-type SOD1 protein (Fig. 5b-ii).

Additionally, the mutant-seeded wild-type filaments showed a different morphology depending on the presence of DTT (Fig. 5b-iii and b-iv). The mutant-seeded wild-type filaments formed in the absence of DTT showed a long and slightly curled morphology (Fig. 5b-iv), whereas the filaments were straight when DTT was present during filament formation (Fig. 5b-iii). However, the overall structures remained curled when DTT was added to the preformed curled wild-type filament, indicating that the disulfide bond is hidden in the filament core (Supplementary Fig. 9e). This finding suggests that intermolecular disulfide formation by Cys57 or Cys146 residues in the wild-type protein might cause a slight curve along the filaments.

**Promotion of SOD1 filament formation by HOCl, but not by H₂O₂.** Our results suggested that the oxidative modification at the Cys57 and Cys146 residues is the triggering factor of the filamentation of the wild-type SOD1. Thus, we next investigated which oxidative stress induces the modification of Cys57 and Cys146 residues and promotes the filament formation of SOD1 protein with the wild-type SOD1 protein. The C6A/C111A mutant SOD1 protein was employed to exclude the modification of non-disulfide forming cysteines and focus on the Cys57 and Cys146, which can form an intramolecular disulfide bond. $H_2O_2$ or HOCl was treated to the wild-type and C6A/C111A SOD1 protein with free thiol forms at Cys57 and Cys146, at various pH and temperature. Then, $H_2O_2$ or HOCl was removed, and the pH was restored to 7.0 by dialysis at 4 °C for the following thioflavin T assay. Notably, we found that the filament formation of the SOD1 protein was promoted by a short exposure (5 min) of the SOD1 protein to 5 mM HOCl at pH 6.4 at 42 °C (Fig. 6a, b). However, the $H_2O_2$ treatment did not promote the filament formation of the wild-type SOD1 protein (Fig. 6a). These observations indicate that the HOCl treatment to wild-type and C6A/C111A SOD1 proteins at a high temperature induced the overoxidation at the cysteine residues, resulting in the promotion of the filament formation.

## Discussion

ALS has been described as a prion-like disease because of its mechanistic similarity; the proteins involved in ALS form pathological filaments which able to replicate and spread by incorporating the normal counterparts, similar to prion proteins[56–61]. Most of the sALS cases caused by filamentous SOD1 occur in people with the wild-type SOD1 gene[62–66]. This study showed that overoxidation of the two cysteine residues in wild-type SOD1 is critical for the structural disturbance of the metal ion binding loop leading to the filamentous form. Of note, the overoxidation-mimicking mutant filaments as seeds promoted the filament formation of the wild-type SOD1 proteins more actively than the wild-type SOD1 filaments themselves. For the seeding effect, filamentous SOD1, especially mutant filaments, may provide molecular platforms with specific interactions with soluble SOD1. In particular, the negative charges at the overoxidized cysteine residues on the molecular platform would provide specific interactions to reduce the activation barrier for the structural transition. We further observed that the pulse treatment of HOCl to the SOD1 protein with the free thiols at the active site under the high temperature and acidic pH condition promoted filament formation like the C57D/C146D SOD1 protein. Since HOCl makes the reactive sulfenyl chloride on cysteine residues[67], the HOCl treatment to the SOD1 protein appears to overoxidize the cysteine residues, unlike $H_2O_2$ mainly making a disulfide bond.

In previous studies, reductive cleavage of the disulfide bond was a prerequisite for filament formation with wild-type SOD1[68–71]. However, the filamentous SOD1 protein was obtained without DTT addition when mutant filaments were added as a seed. This discrepancy can be explained by a reduced SOD1 protein mixed in the wild-type SOD1 protein sample and the high seed effect of the mutant filament. SDS-PAGE analysis under nonreducing conditions revealed that the protein sample of wild-type SOD1 contained the reduced forms in a significant portion (~30%; Supplementary Fig. 1a). Furthermore, the potent seed effect of the mutant filament seemed to facilitate the filament formation on 30% of the reduced form, contained in the wild-type SOD1 sample. We found that the filaments became resistant to dissociation by adding an excess amount of DTT, suggesting irreversible equilibrium kinetics for filament formation in the pool of wild-type SOD1.

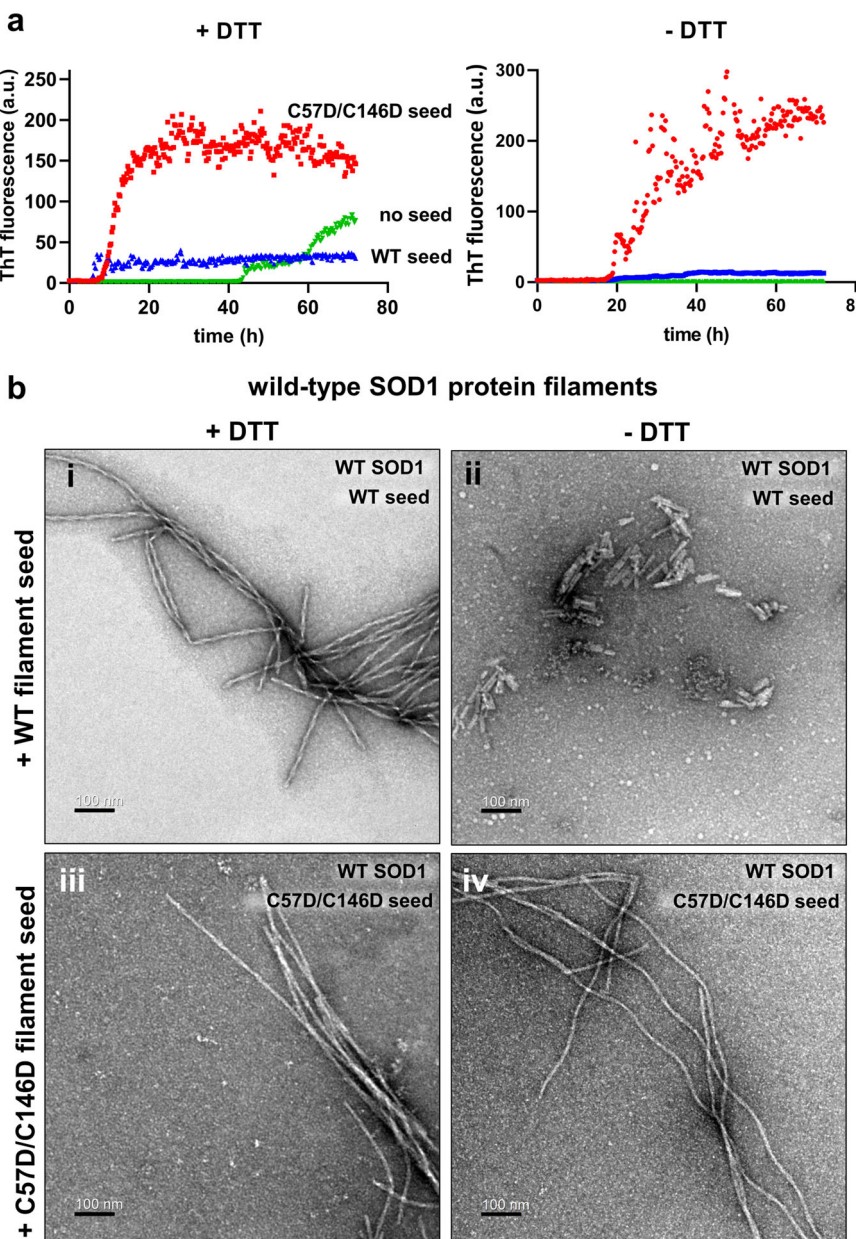

**Fig. 5 Promoted filament formation of the wild-type and C57D/C146D mutant by the preformed filaments. a** ThT fluorescence intensities, representing the filamentation of the wild-type SOD1 protein in the buffer containing wild-type or C57D/C146D mutant SOD1 filament in the presence of 50 mM DTT (left) or the absence of DTT (right). The data points are shown as the mean of three independent experiments. The raw data are presented in Supplementary Fig. 8b. **b** Morphologies of the seeded wild-type SOD1 filaments shown in the negative-stain transmission electron micrographs. The wild-type SOD1 filaments seeded by the wild-type filament in the absence and presence of 50 mM DTT are displayed in (i) and (ii), respectively. The wild-type SOD1 filaments seeded by the C57D/C146D mutant SOD1 filament in the absence and presence of 50 mM DTT are shown in (iii) and (iv). Scale bars indicate 100 nm.

How would wild-type SOD1 be overoxidized in the pathological state? We noted excessive exercise, a well-known risk factor for sporadic ALS[16]. Since transient hypoxia and subsequent ROS surges are caused in the blood and cerebral spinal fluid by excessive exercise, our study noted the reduction and overoxidation of the disulfide bond in the SOD1 protein. The free thiols are oxidized either to disulfide or sulfinic or sulfonic acid in the irreversible overoxidation pathway. In the presence of high ROS stress at a higher temperature, the overoxidation pathway would be preferred. In this study, we further found that the pulsed treatment at acidic pH (pH 6.4) and the high temperature (42 °C) promoted the filament formation in the SOD1 protein with the intact cysteine residues at the active site. The acidic pH and the high temperature can be achievable in the muscle tissue by excessive physical exercise[72–76]. Furthermore, muscle damage by excessive exercise would induce the inflammation response near the neuromuscular junction of the motor neurons. Since the HOCl is generated and secreted by the activated immune cells near the damaged tissues, the local inflammation would increase the likelihood of overoxidizing the SOD1 protein in the motor neuron.

Combined with these findings, we propose a plausible pathway for filament formation in the context of the wild-type SOD1 protein in sALS pathology (Fig. 7). The immature SOD1 protein in the apo-state (apo-SOD1, Fig. 7a) should be processed by

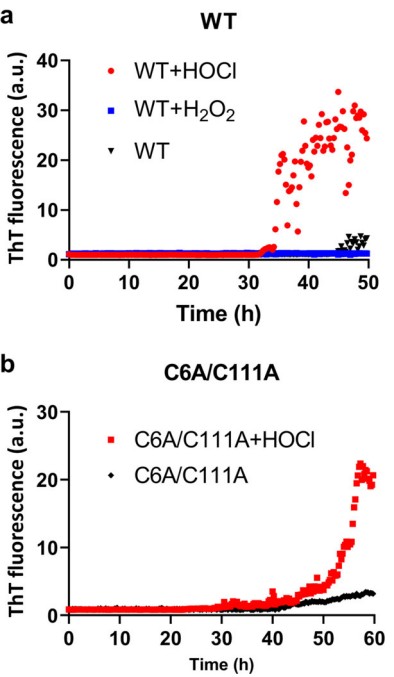

**Fig. 6 Promotion of SOD1 filament formation by HOCl.** The ThT fluorescence intensities represent the filament formation of the wild-type SOD1 protein (**a**) or the C6A/C111A mutant SOD1 protein (**b**). Each protein was treated with HOCl (red), $H_2O_2$ (blue), or none (black). The data points are shown as the mean of three independent experiments. The individual raw datasets are presented in Supplementary Fig. 10.

hCCS in the presence of molecular oxygen and $Cu^{2+}$ ions to be the active SOD1 protein (holo-SOD1, Fig. 7b)[27,36,77]. However, the maturation step would be inhibited in the hypoxic state or without hCCS, increasing the pool of apo-SOD1 in the cytosol. The subsequent local inflammation at a higher temperature would overoxidize the apo-SOD1 proteins. Once the cysteines of SOD1 are overoxidized (ox-SOD1, Fig. 7c), the normal maturation pathway to holo-SOD1 would be inhibited, accumulating ox-SOD1 in the cytosol. Since ox-SOD1 with an open conformation at loop IV would be more prone to be misfolded via monomeric transition, ox-SOD1 would undergo a structural switch to the amyloid-like filamentous form (Fig. 7d). Even at a small amount, the mutant filaments would promote the conversion of the soluble pool of apo-SOD1 to the filamentous form in an irreversible manner. These serial processes accumulate filamentous SOD1, composed of ox-SOD1 and apo-SOD1, leading to the ox-apo-SOD1 filament in the neural cytosol. In the case of repetitive hypoxia and ROS stress in the presence of the local inflammation, ox-SOD1 and apo-SOD1 accumulated, promoting the formation of the ox-apo-SOD1 filament (Fig. 7e). The filaments led to propagation to other regions of the neurons or adjacent neurons (Fig. 7f). In fALS cases, genetic mutations themselves have been shown to destabilize the SOD1 structure, triggering the aggregation or amyloid formation[78,79]. However, we believe that the overoxidation of cysteines contributes to exacerbating the pathogenicity even in the fALS cases, since all SOD1 proteins could be overoxidized by environmental factors.

In this study, we proposed a molecular mechanism by which SOD1 proteins are changed into the filamentous form in ALS pathology, where over-oxidation of disulfide residues may trigger pathogenicity. We noted an ALS risk factor, excessive exercise combined with local inflammation, which is finally linked to the overoxidation of the disulfide bond at the active site. Oxidative stress that is generally increased in sports that demand extreme physical exercises may contribute to recent reports of a significant number of sportsmen with sALS[17,19,20], where over-oxidation of disulfide residues may trigger the pathogenicity. The crystal structure of the overoxidation-mimicking mutant showed structural changes by overoxidation of the disulfide bond at the molecular level. We presented overoxidized SOD1 as the triggering factor of ALS by producing contagious filaments in the propagation of filamentous SOD1 within and between neurons. Our study would help prevent and cure ALS and further provide molecular implications for understanding the other neurodegenerative diseases caused by protein filaments.

## Methods

**Plasmid construction and expression.** The pF151 pcDNA3.1(+)SOD1 WT expressing human SOD1 was purchased from Elizabeth Fisher (Addgene plasmid # 26397; http://n2t.net/addgene:26397; RRID:Addgene_26397). Site-directed mutagenesis was performed to make the expression vector for C57D/C146D mutant SOD1 using the pcDNA3.1-SOD1 plasmid as the PCR template and the primers (5′-GGAGATAATACAGCAGGCGATACCAGTGCAGGTCCTCA-3′, 5′-TGAGGACCTGCACTGGTATCGCCTGCTGTATTATCTCC-3′, 5′-GGAAACGCTGGAAGTCGTTTGGCTGATGGTGTAATTGGGAT-3′, 5′-ATCCCAATTACACCATCAGCCAAACGACTTCCAGCGTTTCC-3′), resulting in pcDNA3.1-SOD1 (C57D/C146D). The *sod1* gene was cloned by PCR and inserted into the pProEx-HTa expression vector (Invitrogen) for bacterial expression by using primers (5′-GGCGAATTCATGGCGACGAAGGCCGTGTGC-3′, 5′-GGCAAGCTTTTATTGGGCGATCCCAATTAC-3′). The resulting pProEx-HTa-SOD1 plasmid encodes the hexahistidine tag and the TEV protease cleavage site at the N-terminus of the protein. Site-directed mutagenesis was performed to make the expression vector for C57D/C146D, C57A/C146A, or C6A/C111A mutant SOD1 using the pProEx-HTa-SOD1 plasmid as the PCR template and the primers (C57D/C146D; described above, C57A/C146A; 5′-GAGATAATACAGCAGGCGCTACCAGTGCAGGTCCTC-3′, 5′-GAGGACCTGCACTGGTAGCGCCTGCTGTATTATCTC-3′, 5′-GAAACGCTGGAAGTCGTTTGGCTGCTGGTGTAATTGGGA-3′, 5′-TCCCAATTACACCAGCAGCCAAACGACTTCCAGCGTTTC-3′, C6A/C111A; 5′-GACGAAGGCCGTGGCCGTGCTGAAGGGC-3′, 5′-GCCCTTCAGCACGGCCACGGCCTTCGTC-3′, 5′-CACTCTCAGGAGACCATGCCATCATTGGCCGCACAC-3′, 5′-GTGTGCGGCCAATGATGGCATGGTCTCCTGAGAGTG-3′), resulting in pProEx-HTa-SOD1 (C57D/C146D), pProEx-HTa-SOD1 (C57A/C146A), and pProEx-HTa-SOD1 (C6A/C111A). The plasmids pProEx-HTa-SOD1, pProEx-HTa-SOD1 (C57D/C146D), pProEx-HTa-SOD1 (C57A/C146A) and pProEx-HTa-SOD1 (C6A/C111A) were transformed into *E. coli* BL21 (DE3). The transformed cells were cultured in 1.5 l of Luria Broth medium containing 100 μg/ml ampicillin and 34 μg/ml chloramphenicol at 37 °C until an $OD_{600}$ of 0.8 was measured. Protein expression was induced with 0.5 mM isopropyl 1-thio-β-D-galactopyranoside at 30 °C for 6 h with supplementation of 0.1 mM $CuCl_2$ and 0.1 mM $ZnCl_2$.

**Protein purification.** The cells were harvested by centrifugation, and then the cell pellets were resuspended in lysis buffer containing 20 mM Tris-HCl (pH 8.0), 150 mM NaCl, and 2 mM 2-mercaptoethanol. The cell suspensions were disrupted using a continuous-type French press (Constant Systems Ltd., United Kingdom) at 23,000 psi. After centrifugation at 19,000 g for 30 min at 4 °C, the supernatant was loaded onto a nickel-nitrilotriacetic acid column (GE Healthcare). The column was washed with ~300 ml of lysis buffer supplemented with 20 mM imidazole. Then, the SOD1 proteins were eluted with lysis buffer supplemented with 250 mM imidazole. The eluted fractions were treated with TEV protease to cleave the hexahistidine tag. The tag-free proteins were incubated with 0.05 mM EDTA for 30 min, followed by 0.1 mM $CuCl_2$ for 30 min at 4 °C. The protein sample was diluted fourfold with 20 mM Tris-HCl (pH 8.0) buffer and loaded onto a HiTrap Q column (GE Healthcare). A linear gradient of increasing NaCl concentrations was applied to the column. The fractions containing the protein were pooled, concentrated, and subjected to size-exclusion chromatography using a HiLoad 16/600 Superdex 200 pg column (GE Healthcare) in 20 mM Tris-HCl (pH 8.0) buffer containing 150 mM NaCl. The resulting protein was concentrated to 8 mg/ml and stored at −80 °C until use for crystallization. Protein purity was checked on a 15% polyacrylamide gel, and the concentration was determined by optical spectroscopy.

**Crystallization of SOD1.** The wild-type SOD1 proteins were crystallized at 16 °C using the hanging drop vapor diffusion method after mixing 1 μl protein solution and 1 μl well solution containing 0.1 M sodium acetate-HCl (pH 4.5) and 21% (w/v) PEG 3350. The C57D/C146D SOD1 protein was crystallized under the same conditions except for the mother solution, which contained 0.2 M $MgCl_2$ and 22% (w/v) PEG 3350. The crystals were briefly incubated in the mother solution supplemented with 25% (v/v) DMSO as cryoprotectants and flash-frozen in liquid nitrogen to collect the datasets.

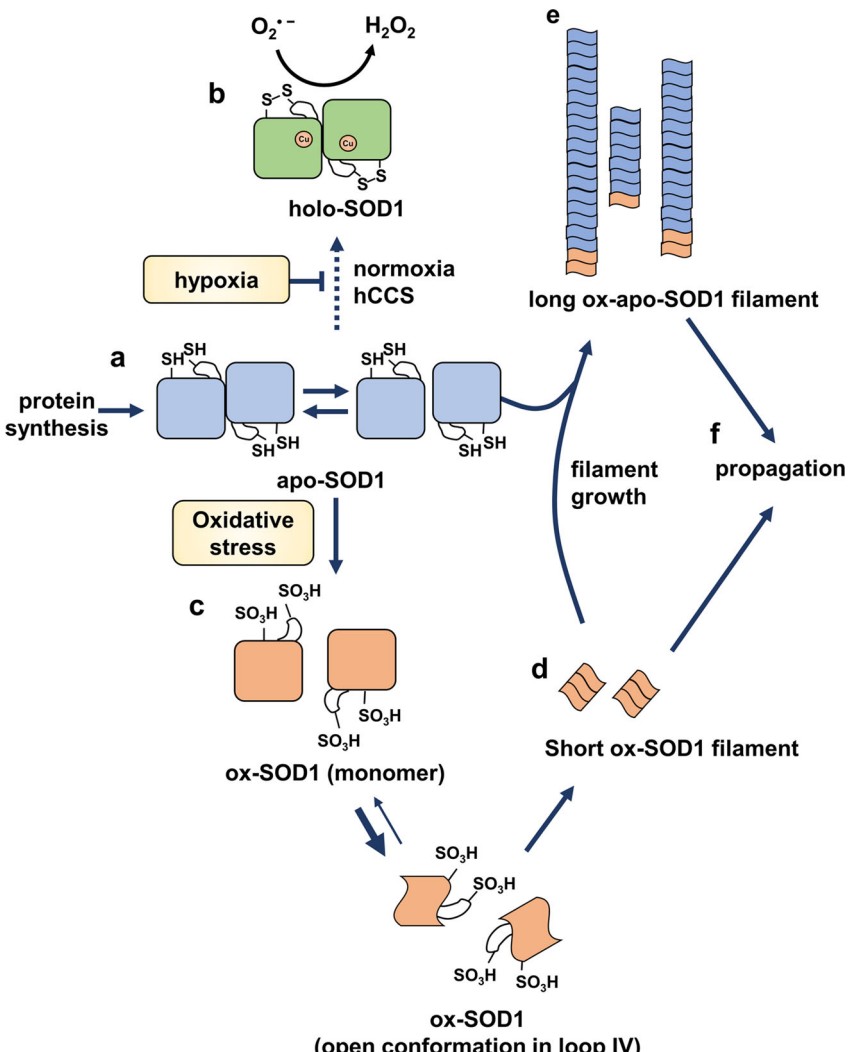

**Fig. 7 Proposed mechanism of sALS development from wild-type SOD1. a** Newly synthesized SOD1 does not have Cu$^{2+}$, and the cysteine residues are reduced, resulting in apo-SOD1 in an equilibrium between the monomeric and dimeric forms. **b** In the normoxic state, apo-SOD1 undergoes the maturation process. Disulfide bond formation and copper insertion occur by hCCS, resulting in holo-SOD1 with full superoxide dismutase activity. Under hypoxic conditions, disulfide bond formation by hCCS is inhibited, and immature apo-SOD1 accumulates in the cytosol. **c** In the presence of excess ROS, such as HOCl, with unknown environmental factors, a small amount of apo-SOD1 is overoxidized in an irreversible manner, resulting in ox-SOD1. **d** The structure of Ox-SOD1 with the open conformation in loop IV starts to form a short filamentous structure. **e** Filamentous ox-SOD1 is rapidly elongated by recruiting apo-SOD1, resulting in the ox-apo-SOD1 filament. **f** Filamentous ox-apo-SOD1 propagates to other regions in neurons or adjacent neuronal cells.

**Data collection and structural determination**. Diffraction datasets of the wild-type crystals were collected at the Pohang Accelerator Laboratory Beamlines 5 C and 11 C in the Republic of Korea[80]. The datasets were processed using the HKL-2000 package[81]. The structures of the wild-type and mutant SOD1 proteins were determined using the molecular replacement method with MOLREP in the CCP4 package[82] using a human SOD1 structure (PDB ID: 1PU0) as a search model[42]. All the structures were built using COOT and refined using PHENIX refine software[83,84]. The detailed statistics are shown in Table 2.

**Analysis of the metal contents of the purified SOD1 proteins**. The metal contents of the SOD1 proteins were determined using inductively coupled plasma mass spectrometry (ICP-MS) at the mass spectrometry facility of the Seoul National University National Instrumentation Center for Environmental Management (SNU NICEM). The EPA method 3051[85] was used for sample processing by microwave-assisted acid digestion.

**Superoxide dismutase activity assay**. The enzymatic activity of SOD1 was measured in a 100 µl reaction buffer containing 50 mM sodium phosphate (pH 7.4), 0.2 mM NADH, 0.2 mM cytochrome $c$, and 240 µg/ml $E.\ coli$ C43A/C48A mutant RclA protein as NADH oxidase[37]. The wild-type, C57D/C146D mutant, or C57A/C146A mutant SOD1 protein was added to the reaction buffer at a concentration of 32 µg/ml. Cytochrome $c$ from the equine heart was purchased from Sigma-Aldrich,

and the recombinant C43A/C48A mutant RclA was produced as previously reported[37]. The reaction mixtures were incubated in the clear-walled, flat bottom, 96-well plates (SPL Life Sciences, Korea) for 5 min at 25 °C. The increase in the absorbance at $\lambda = 550$ nm by cytochrome $c$ reduction by superoxide anion was recorded every 10 s using a Varioskan Lux Multimode microplate reader (Thermo Fisher Scientific, USA). The initial superoxide production rates were calculated from the reaction for the initial 30 s in six independent experiments. The inhibition rate of superoxide production by SOD1 (% inhibition) was derived by the following equation based on previous research[86].

$$\%\text{inhibition} = (1 - \text{uninhibited O}_2^{\bullet-} \text{ production rate/inhibited O}_2^{\bullet-} \text{ production rate}) \times 100\%$$

**Size-exclusion chromatography coupled with multiangle light scattering (SEC-MALS)**. The freshly thawed wild-type or C57D/C146D mutant SOD1 protein in buffer containing 20 mM Tris (pH 8.0) and 150 mM NaCl was used for the SEC-MALS analysis. Each sample was subjected to size-exclusion chromatography (SEC) on a Superdex 200 increase 10/300 GL column (GE Healthcare). The molecular sizes and oligomerization states of SOD1 were measured by MALS (DAWN HELIOS II; Wyatt Technology, USA).

**Tm measurement by thermal shift assay**. Thermal shift assays were performed on the Bio-Rad CFX Connect$^{\text{TM}}$ Real-Time System (Bio-Rad Laboratories, Hercules, CA,

**Table 2 Data collection and refinement statistics.**

|  | Wild-type SOD1 | C57D/C146D mutant SOD1 |
|---|---|---|
| **Data collection** | | |
| Beamline | PAL 11 C | PAL 5 C |
| Wavelength (Å) | 0.97942 | 0.97949 |
| Space group | $P3_2$ | $P2_1$ |
| Cell dimensions | | |
| a, b, c (Å) | 112.430, 112.430, 209.280 | 49.319, 48.554, 55.015 |
| α, β, γ (°) | 90, 90, 120 | 90, 93.17, 90 |
| Resolution (Å) | 50.0–2.70 (2.75–2.70) | 50.0–1.80 (1.83–1.80) |
| $R_{pim}$ | 0.067 (0.157) | 0.045 (0.139) |
| $R_{merge}$ | 0.167 (0.294) | 0.113 (0.297) |
| $I/\sigma$ | 8.3 (3.08) | 12.0 (3.8) |
| Completeness (%) | 93.8 (78.9) | 96.7 (84.2) |
| Redundancy | 4.9 (2.5) | 4.8 (3.3) |
| **Refinement** | | |
| Resolution (Å) | 49.52–2.7 | 36.38–1.80 |
| No. reflections | 74868 | 21300 |
| $R_{work}/R_{free}$ | 0.2303/0.2473 | 0.1843/0.2130 |
| No. total atoms | 14044 | 2363 |
| Wilson B-factor (Å$^2$) | 41.74 | 16.95 |
| RMS deviations | | |
| Bond lengths (Å) | 0.001 | 0.008 |
| Bond angles (°) | 0.347 | 1.104 |
| Ramachandran plot | | |
| Favored (%) | 94.40 | 98.95 |
| Allowed (%) | 5.44 | 1.05 |
| Outliers (%) | 0.16 | 0.0 |
| PDB ID | 7FB9 | 7FB6 |

USA) using a Protein Thermal Shift$^{TM}$ Dye Kit (Thermo Fisher Scientific, USA). SOD1 proteins (30 μM) in 20 mM Tris (pH 8.0) buffer containing 100 mM NaCl were used in the thermal shift assays. The temperature was measured from 4 °C to 95 °C at intervals of 1 °C to calculate the Tm value of proteins. Data were collected as three individual endpoint readings for each 1 min cycle. The predefined experimental method of the CFX instrument was chosen to enable software calculation of the first derivative values for the denaturation curve raw data points. The melting temperature of a protein (Tm, the temperature at which there is 50% denaturation) was determined by the temperature at the maximum velocity of the fluorescence curve, which is the temperature at the local minimum point of first derivatives [-d(fluorescence)/dT].

**Protein proteolysis using trypsin**. The SOD1 proteins (0.5 mg/ml) were incubated in 100 μl of 50 mM sodium phosphate (pH 7.4) buffer containing 150 mM NaCl, 10 μg/ml trypsin at 37 °C for 30 min or 1 h in the presence or absence of 5 mM DTT. The reactions were stopped by adding 6X Laemmli sample buffer and immediately boiling the sample at 100 °C before loading for SDS-PAGE. The resulting proteins were detected by Coomassie Blue staining.

**Circular dichroism and protein secondary structure analysis**. Far-UV CD spectra of the wild-type and C57D/C146D mutant SOD1 proteins (0.6 mg/ml) were recorded in 50 mM sodium phosphate (pH 7.4) buffer at 25 °C using a 1 mm pathlength cell on a Jasco J-1500 spectropolarimeter (Korea Basic Science Institute, Ochang, Republic of Korea). The protein secondary structure was estimated by Spectra Manager Version 2 Protein Secondary Structure Estimation program version 2.01.00 (JASCO Corporation).

**Cell culture and transfection**. SK-N-SH cells were purchased from Korean Cell Line Bank (KCLB, Seoul, South Korea) and maintained in MEM medium containing 10% fetal bovine serum, 1% antibiotics, 25 mM HEPES, 300 mg/L L-Glu, and 1% penicillin-streptomycin DMEM at 37 °C and 5% CO$_2$/95% air with humidification. Zn$^{2+}$ scavenger (TPEN; P4413) was obtained from Sigma-Aldrich (St, Louis, Mo, USA). Cu$^{2+}$ scavenger (ATN-224; CAS 649749-10-0) was purchased from Cayman Chemical (Michigan, USA). SK-N-SH cells were seeded on glass coverslips, and wild-type or its C57D/C146D variant in the pcDNA3.1 vector was transfected into SK-N-SH cells and incubated with 1 μM TPEN or 10 μM ATN-224 for 24 h.

**Dot blot analysis**. Transfected cells with the SOD1 expression vectors (WT, C57D/C146D in pCDNA3.1) were treated with chemicals (1 μM TPEN, 10 μM ATN-224). After 24 h, cells were lysed with TNN buffer (50 mM Tris-Cl, pH 7.5, 150 mM NaCl, 0.3% NP-40) not containing detergent, and then the cell lysates were immobilized on a nitrocellulose membrane using the Bio-Dot SF Microfiltration apparatus (Bio-Rad Laboratories, Hercules, CA). Each membrane was washed with TBS and blocked by 5% BSA to remove the background for 1 h. After blocking, the membrane was incubated with the mouse monoclonal anti-misfolded SOD1 (B8H10) antibody (1:3,000, MediMabs, # MM-0070-P) or the anti-actin antibody (1:5,000 in 1% BSA blocking buffer, Proteintech, #66009-1-Ig) for 2 h, and then reacted with secondary antibody (goat anti-mouse IgG-horseradish peroxidase, 1:50,000 in 1% BSA blocking buffer, Thermo Fisher Scientific, #31430) for 40 min. The antibody binding was detected with ECL and X-ray film exposure. Actin was used as the loading control.

**Immunofluorescence**. The transfected cells on coverslips were fixed with 4% paraformaldehyde for 30 min at room temperature. The cells were then permeabilized in 0.1% Triton X-100/PBS for 10 min, followed by 1 h of treatment with a blocking solution consisting of normal goat serum (1:200, Abcam, ab7481) in PBS. The pan-SOD1 antibody (1:400, Genetex; GTX100554) solution, diluted in the blocking solution, was added to the cell solution overnight at 4 °C. After washing with PBS, we incubated the cells with Alexa Fluor 488 antibody (Goat anti-Rabbit IgG, 1:500, Thermo Fisher Scientific, #A-11008) at 4 °C for 6 h. The nucleus was stained with 4,6-diamidino-2-phenylindole (DAPI), and the endoplasmic reticulum (ER) was stained using ER-Tracker$^{TM}$ Red dye (Invitrogen) for 10 min. After washing the coverslips three times with PBS, we mounted the coverslips with mounting solution (H-5501; Vector Laboratories) for fluorescence microscopy (Zeiss).

For inclusion positive cells counting, immunofluorescence images with SOD1 antibodies were counted in randomly selected fields. Depending on SOD1 staining, inclusion positive cells were counted based on the strong intensity of SOD1 and expressed as a percentage of total cells counted.

**Filament formation of the SOD1 proteins**. Purified recombinant SOD1 proteins (50 μM as a monomer) were in 100 μl of 30 mM sodium phosphate (pH 7.4) buffer containing 165 mM NaCl, 10 mM EDTA, 20 μM thioflavin T, and various concentrations of DTT (0 mM to 100 mM). The reaction samples were plated onto Corning® 96-well black polystyrene clear bottom microplates (CLS3603 Sigma–Aldrich, St. Louis, MO, USA) and covered by a clear adhesive film. The 96-well plates were incubated with pulsed shaking at 480 rpm for 1 min at the start of a 2 min cycle at 37 °C in a Varioskan Lux Multimode microplate reader (Thermo Fisher Scientific, USA). The ThT fluorescence was measured in a microplate reader at an excitation wavelength of 440 nm and an emission wavelength of 485 nm every 15 min. The ThT data were plotted as the mean of the ThT fluorescence readings (in arbitrary units, a.u.) from triplicate or sextuplicate experiments, and then the lag time and the increase in fluorescence values were calculated from the plot. For the seeded filament formation, filament seeds were added to the reaction mixture at 5% (w/w) of the total protein amount in each assay before incubation in the microplate reader.

For the filament formation of ROS-treated SOD1, purified recombinant wild-type or C6A/C111A mutant SOD1 proteins (200 μM as a monomer) in 200 μl of 100 mM sodium phosphate buffer pH 6.4 containing 150 mM NaCl were preheated at 42 °C. After 10 min of incubation, 5 mM of HOCl or H$_2$O$_2$ was added to the protein solution and incubated at 42 °C for another 5 min. The ROS-treated protein solution was then dialyzed with 500 ml of 50 mM sodium phosphate (pH 7.0) buffer containing 150 mM NaCl for 3 h by 2 times. The dialyzed proteins were used for filament formation assay without reducing agents. Control assay was performed using untreated SOD1 proteins.

**Analysis of the SOD1 filament by transmission electron microscopy (TEM)**. Ten microliters of the reaction mixture containing the filamentation of SOD1 proteins at the end stage was applied on carbon-coated 400-mesh copper grids, freshly glow-discharged. The grids were washed three times with 100 μl distilled water after 1 min incubation. Then, the grids were stained with 1% (w/v) uranyl acetate, blotted with filter paper after 1 min, and allowed to air dry. Images were recorded using an energy-filtering transmission electron microscope (EFTEM), Carl Zeiss LIBRA 120 (Carl Zeiss, Oberkochen, Germany), with an acceleration voltage of 120 kV.

**Statistics and reproducibility**. The sample size and number of replicates for each experiment were described in methods and figure legends. GraphPad Prism 8.0.1 was used for statistical analysis. Comparisons between two groups were analyzed by the 2-tailed Student's $t$-test. Data were presented as the mean ± SD, and $p$-value < 0.05 is considered significant.

**Reporting summary**. Further information on research design is available in the Nature Research Reporting Summary linked to this article.

## Data availability

The atomic coordinates and structure factors are deposited into the RCSB Protein Data Bank (http://www.pdb.org), respectively: wild-type SOD1 (PDB:7FB9) and C57D/C146D

mutant SOD1 (PDB: 7FB6). The newly generated plasmids are deposited in Addgene, respectively: pcDNA3.1-SOD1 C57D/C146D (Addgene ID: 191848), pProEx-HTa-SOD1 WT (Addgene ID: 191849), pProEx-HTa-SOD1 C57A/C146A (Addgene ID: 191850), pProEx-HTa-SOD1 C57D/C146D (Addgene ID: 191851), and pProEx-HTa-SOD1 C6A/C111A (Addgene ID: 191852). The uncropped and unedited blot/gel images for the main figures are included in Supplementary Figure 11. The numerical source data for graphs in the main figures are included in Supplementary Data 1. All other data are available from the corresponding author on reasonable request.

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

## Acknowledgements

We would like to thank the Pohang Accelerator Laboratory (Pohang, Republic of Korea) for its Beamline 5 C and 11 C equipment. We also thank the SEC-MALS and CD facility at the Korea Basic Science Institute (Ochang, Republic of Korea). This work was supported by grants from the National Research Foundation of Korea (NRF) funded by the MSIT (2022R1A2C109178311 to NCH) and the Basic Research Program through the National Research Foundation of Korea (NRF) funded by the MSIT (2020R1A4A1019322 to NCH). This research was also supported by Korea Institute of Planning and Evaluation for Technology in Food, Agriculture, Forestry and Fisheries (IPET), funded by Ministry of Agriculture, Food and Rural Affairs (MAFRA) (321036052HD020).

## Author contributions

Y.B. and N.-C.H. conceived and designed the study. Y.B., T.-G.W., and D.L. performed experiments. Y.B., J.A. and T.-G.W. analyzed the data. Y.K. provided materials. Y.K., B.-J.P. and N.-C.H. supervised the research. Y.B., T.-G.W., and N.-C.H. wrote the paper.

## Competing interests
The authors declare no competing interests.
