## [Peer Review File · Communications Biology]

Reviewers' comments:

Reviewer #1 (Remarks to the Author):

SOD1 was the first protein molecule that was directly linked to amyotrophic lateral sclerosis (ALS) some 30 years ago. Evidence has emerged that it is involved in some of the sporadic ALS also making the understanding of the pathogenic mechanism highly important. For mutant SOD1 it has been shown that mutant themselves may destabilise the molecule triggering the aggregation/amyloid formation. However, for sALS where wild type SOD1 is presumed to be involve it the trigger for amyloid remains to be determined. Hypoxic stress on the cells has been shown to induce a disulphide reduction. It is also known that disulphide forming Cys146 is prone to be overoxidised.

Here, the authors explore the molecular mechanism of SOD1 filament formation by cysteine overoxidation in sporadic ALS (sALS). For this they have created a double mutant (Cys57Asp/Cys146Asp) to mimick the overoxidation of disulfide-forming cysteine residues. The study is well conducted and provides a significant advance towards a possible mechanism where reactive oxygen species may play a role (in combination with other factors such as metal incorporation etc references 40-42). I recommend publication of the manuscript with minor revision:

1. In the abstract and introduction, the creation of double mutant (Cys57Asp/Cys146Asp) to mimick overoxidation be explicitly stated.
2. In discussion, a clear statement should be included to state that in familial ALS, destabilisation of mutant SOD1 caused by the mutation themselves has been shown to destabilise the molecule triggering the aggregation/amyloid formation with appropriate references (e.g. Nature Structural & Molecular Biology 10, 461-467 & PNAS 101, 5976-5981), where overoxidation may further exacerbate the pathogenicity.
3. In concluding remarks it would be desirable to add "Oxidative stress that is generally increased in sports that demand extreme physical exercises may contribute to recent reports of a significant number of sportsmen with sALS, where over-oxidation of disulphide residues may trigger the pathogenicity". In this context, authors may wish to look at <https://www.bbc.co.uk/news/uk-england-leeds-60998381> and other related stories.

Reviewer #2 (Remarks to the Author):

There are two basic questions for this study. First one is whether C57D/C146D mutation surely mimic to the overoxidized Cys57 and Cys146 nor not. Schematic presentations of the cysteine-sulfenic or sulfonic acid forms should be added in figure for the readers. Second one is why the authors didn't initially compare with the previous study of C57A/C146A?. The reference 43 must be very important for this study (no citation in "Introduction"), because of the similar strategy for site-directed mutagenesis of Cys57 and Cys146. The authors should also prepare this mutant originally, and compare with C57D/C146D mutant in parallel.

Other suggestions are as follows.

1. L99, whether protein conformation is elucidated based on the peak shape of gel-filtration?
2. L107, Can the authors calculate the content of secondary structure based on CD spectrum?
3. Crystallization conditions for the WT and mutant proteins are significantly different. Is this related to their conformational change?
4. L215-222, It is likely that data for C6/C111A mutant is not necessary.
5. First and second paragraph in "Discussion" are completely overlapped with results sections.
6. There are much numbers of sentences with "We" as a subject...

Reviewer #3 (Remarks to the Author):

The authors in the given manuscript have performed studies to describe the role of over-oxidized

cysteines (C57/C146) that form the intra-disulfide bond in sporadic ALS (sALS). Although the authors have done significant experimental work to prove the contribution of C57/C146 residues in sALS, the manuscript has some shortcomings that the authors need to address.

1. Figure 1B – The color description in the SEC-MALS figure is confusing. The authors should clarify the legends carefully to avoid ambiguity.
2. "...demonstrated that the C57D/C146D mutant SOD1 protein was mostly monomeric in solution..." The SEC graph shows multiple populations for the C57/C146 mutant (red line), monomer, dimer, and aggregate formation as well? How do the authors justify the correlation with MALS where it shows it is monomeric form only? Please explain in detail. The figure should be individually plotted for SEC and MALS if there exists no correlation?
3. Please mention if the SEC-MALS experiment was carried out using a freshly thawed protein or a time/temperature-incubated protein? The authors are required to furnish details on the buffer and incubation terms used in the methodology section of SEC-MALS.
4. During the C57D/C146D mutant purification steps, does the final gel filtration chromatogram show more than one peak or elution corresponding to a higher molecular weight? The authors requested to justify using the purification SEC chromatogram.
5. "...the monomeric feature of the mutant SOD1 is different from the disulfide-reduced wild-type SOD1." This sentence questions the use of the mutated C57D/C146D form of SOD1. Please justify.
6. CD spectra do not show too much deviation from the control. Does it justify the fibril formation due to the open conformation of the C57D/C146D mutant?
7. Why was the thermal shift assay carried out in a different buffer system than the one it was stored in? If it has buffer-induced variations in thermal stability, are they not affected during the aggregation experiments or storage of the protein itself? Please justify.
8. "...structural features distinct from the reduced form of wild-type SOD1 with free thiols at Cys57 and Cys146." How do the authors confirm that the wild type has free thiols at C57 and C146 at 5 mM DTT? The previous results in Fig 1 suggest that the metal ions are intact, and there is no proteolysis at all due to the reduction of thiols at the above residues? Doesn't the reduction of an intra-disulfide bridge require higher DTT? Also, at 5 mM DTT, if the residues are at free thiols, it should behave similarly to the C57D/C146D mutant. Please justify.
9. There is the usage of different concentrations of DTT throughout the work. Please justify the usage of the different concentrations of DTT in each of the experiments when used.
10. The spelling of ATN-224 in the text should be corrected.
11. The statistical Significance value should be calculated and mentioned in Figure 1F.
12. Figure 2A – Justify the presence of SOD1 monomers for pcDNA3.1 and SOD1 wild-type.
13. Figure 2A - The authors are required to provide quantification and significance, implying statistics in the Western blot figure.
14. "...the overoxidation mimicking C57D/C146D mutant SOD1 exhibits a higher propensity to form filaments in the cellular environment." – The lane corresponding to this is not showing the formation of higher aggregates. Please justify.
15. Figure 2A – Based on the results shown in the previous Figure 1A, the WB blot should be exhibiting higher aggregates, irrespective of the presence or absence of the metal chelators. Please justify.
16. The results for Figure 2A are required to be explained in detail in the manuscript.
17. Previous studies specify that the loss of metals through chelation causes aggregation-prone effects. Please justify why the Western blotting results do not show any for wild-type?
18. The authors have used the pan-SOD1 antibody to assess the formation of aggregates and not a misfolded SOD1 antibody. The authors are required to reassess the WB results with the misfolded SOD1 antibody.
19. Figures 2A and 2B do not correlate for wild-type or C57D/C146D mutant in the presence of ATN-224. Please justify.
20. Statistical quantification is required for the immunofluorescence images. The authors are requested to write in detail the same in the text as well.
21. Supplementary Figure 3 – Please mention which map (2Fo-Fc map or difference Fourier map or Omit map) is shown here (legend and text).

22. Figure 3A – The hCCS is marked wrongly in the figure.
23. Electron density (Omit map or Polder map) of the copper and zinc ions is required to be shown for the C57D/C146D structure.
24. Supplementary Fig 4A – The superposition of the two models should be done with one chain to see dimeric angle variations as well.
25. “ ... structure to the wild-type SOD1 with an RMSD of 0.710 Å between 152 C α atoms (Supplementary Fig. 4A) (residues 2-253).”
The rmsd value is too small for such a large structural difference in the loop regions (Figure 3B and Suppl Figure 4A). How was the superposition performed? Performed with respect to one chain (A or B) or both? Performed on the structural region only with respect to one chain (A or B) or both? Write in detail.
26. “On the structural superposition with the wild-type dimeric structure...” the superposition indicates loss of G51 interaction but with a missing loop density. Figure 3A shows no missing loop in the second chain. The inference is invalid as the succeeding missing loop makes it ambiguous.
27. Supplementary Figure 4C – The side chain electron density of R144 is absent. Again, which electron density have shown here? The conformations of the side-chain of R144 in both the chains cannot be accounted for as there is no density in these regions. In the PDB file, the occupancy corresponding to these atoms was set to ‘zero’? Conclusions cannot be inferred from the missing density to evaluate disengagement of the residue, which is proposed to lead to superoxide catalytic activity deactivation. A biochemical assay on the R144 mutant is required to validate the inference.
28. It is advised to the authors to maintain the same chain colors for Supp. Fig 4 and Fig. 3 maintain a clear understanding of the structures.
29. Data analysis of ThT measurements should be included in the methodology.
30. Was the ThT setup performed for three individual experimental replicates or triplicate within the same plate?
31. Why is there a difference in the ThT aggregation times without and with DTT for the C57D/C146D over-oxidation mutant?? Justify.
32. Why were a different pre-incubation and buffer conditions used for the C6/C111 mutant as compared to the major C57D/C146D mutant? The ThT graphs for the C6/C111 mutant in comparison with the wild-type should also be shown. If both the mutants are meant to cleave intra-disulfide bonds, the condition of amyloid formation assessment should not vary. The authors need to reassess the experimental conditions used for the ThT analysis with C6-C111 mutant.

Point-by-point response to reviewers' comments

Reviewer #1

Comment 1: In the abstract and introduction, the creation of double mutant (Cys57Asp/Cys146Asp) to mimic overoxidation be explicitly stated.

Response: We stated the creation of double mutant (Cys57Asp/Cys146Asp) to mimic overoxidation in the abstract (**p1, line 17-19**) and introduction (**p4, line 63-72,**). We also added schematic presentations of the cysteine-sulfinic or sulfonic acid forms in the **Fig. 1A and Supplementary Fig.1.**

Comment 2: In discussion, a clear statement should be included to state that in familial ALS, destabilization of mutant SOD1 caused by the mutation themselves has been shown to destabilize the molecule triggering the aggregation/amyloid formation with appropriate references (e.g. Nature Structural & Molecular Biology 10, 461-467 & PNAS 101, 5976-5981), where overoxidation may further exacerbate the pathogenicity.

Response: As the reviewer suggested, we added the statement at the end of the discussion (**p16, line 355-359**).

Comment 3: In concluding remarks it would be desirable to add “Oxidative stress that is generally increased in sports that demand extreme physical exercises may contribute to recent reports of a significant number of sportsmen with sALS, where over-oxidation of disulfide residues may trigger the pathogenicity”. In this context, authors may wish to look at <https://www.bbc.co.uk/news/uk-england-leeds-60998381> and other related stories.

Response: We restated the statement of conclusion by including the statement recommended by the reviewer (**p16, line 361-364**)

Reviewer #2

Comment 1: Whether C57D/C146D mutation surely mimic to the overoxidized Cys57 and Cys146 nor not. Schematic presentations of the cysteine-sulfenic or sulfonic acid forms should be added in figure for the readers.

Response: We added schematic presentations of the cysteine-sulfenic or sulfonic acid forms in the **Fig. 1A and supplementary Fig. 1**.

Comment 2: Why the authors didn't initially compare with the previous study of C57A/C146A? The reference 43 must be very important for this study (no citation in "Introduction"), because of the similar strategy for site-directed mutagenesis of Cys57 and Cys146. The authors should also prepare this mutant originally, and compare with C57D/C146D mutant in parallel.

Response: We prepared C57A/C146A mutant SOD1 originally and compared the filament formation ability with C57D/C146D mutant SOD1. (**Fig. 4A**) (**p10-11, line 214-217**) The filament formation by C57A/C146A mutant SOD1 was faster than wild-type SOD1 and slower than the C57D/C146D mutant SOD1. This supports that the overoxidation of disulfide-forming cysteines makes the SOD1 to be more disordered and prone to filament formation than the mere disulfide breakage.

We also added biochemical data to comparing the C57D/C146D mutant with the C57A/C146A mutant SOD1. (Fig. 1C, 1F and supplementary Fig. 2D) (p6-7, line 108-113, 123-126, 127-136)

Comment 3: L99, whether protein conformation is elucidated based on the peak shape of gel-filtration?

Response: The open conformation of the mutant protein was elucidated primarily based on the peak shape of gel-filtration, as we commented in the manuscript. As the reviewer noticed, the peak shape of the mutant protein was broad. Moreover, we also considered the elution volume of the mutant protein in the gel-filtration. We noted that the monomeric mutant protein was eluted faster than the dimeric wild-type protein, which indicated that the mutant protein behaved as bulkier or more open in the solution than the wild-protein, as was commented in the revised manuscript. (p5, line 97-100)

Comment 4: L107, Can the authors calculate the content of secondary structure based on CD spectrum?

Response: We already had the content of the secondary structure based on the CD spectrum. Thus, we added the calculated content in the revised manuscript (**Fig. 1D**).

Comment 5: Crystallization conditions for the WT and mutant proteins are significantly different. Is this related to their conformational change?

Response: We obtained the crystals of the wild-type SOD1 and C57D/C146D mutant SOD1 at different pH. The wild-type SOD1 was crystallized at pH 4.5, and the C57D/C146D mutant SOD1 was at pH 8.0. However, the previous structures of the wild-type SOD1 crystallized at pH 7.5 or 8.0 (PDB ID: 1PU0 and 2C9U) were highly similar to the structure of wild-type SOD1 at pH 4.5 in this study. Thus, the conformational change between the wild-type and mutant SOD1 seemed independent of the pH in the crystallization solution. Therefore, we believe that the conformational changes between the wild-type and mutant proteins result from the mutation effect.

Comment 6: L215-222, It is likely that data for C6/C111A mutant is not necessary.

Response: The data (Fig. 4B in original manuscript) and descriptions (line 215-222 in original manuscript) for C6A/C111A mutant SOD1 was removed with the feedback from the reviewer.

Comment 7: First and second paragraph in “Discussion” are completely overlapped with results sections.

Response: We removed the redundant descriptions in the first and second paragraphs in Discussion. (p14, line 301-316)

Comment 8: There are much numbers of sentences with “We” as a subject...

Response: We changed sentences and removed the unnecessary “We” from the manuscript. (line 83-84, 94-95, 109-110, 127-128, 140, 141, 147, 165-166, 210, 249-251, 251-252, 266-267, 275-276, 290, 318)

Reviewer #3

Comment 1: Figure 1B – The color description in the SEC-MALS figure is confusing. The authors should clarify the legends carefully to avoid ambiguity.

Response: We added a legend in **Figure 1B** for the clarification of the graph. We also updated the figure legend to clarify the description of Figure 1B. (p 35, line 809-812)

Comment 2: "...demonstrated that the C57D/C146D mutant SOD1 protein was mostly monomeric in solution..." The SEC graph shows multiple populations for the C57/C146 mutant (red line), monomer, dimer, and aggregate formation as well? How do the authors justify the correlation with MALS where it shows it is monomeric form only? Please explain in detail. The figure should be individually plotted for SEC and MALS if there exists no correlation?

Response:

We performed the SEC-MALS (SEC combined with MALS) with the purified SOD1 proteins (wild-type and the C57D/C146D mutant). Since the MALS results were from each elution fraction of the SEC, the SEC and MALS must be correlated, and two results should be combined for the correct interpretation.

We employed three independent methods (the UV absorption, RI absorption, and light scattering (LS)) to better measure the concentrations of the proteins, which is important in precise measurement of the molecular weight of the samples. As the reviewer noticed from the elution profiles, the mutant protein showed a broader peak than the wild-type protein in a sharp peak. Probably, the reviewer thought that the C57D/C146D mutant proteins consist of the multiple populations of the monomer, dimer, and aggregate form. However, the MALS results of the peak indicated that all the populations of the mutant protein were only in the monomeric

form (~16 kDa) except for a small portion at the start of the peak. We eliminated the second peak from the plot in this revision since the peak did not correspond to the SOD1 protein (see Response to Comment 4 for the detail). (**Fig. 1B**)

Comment 3: Please mention if the SEC-MALS experiment was carried out using a freshly thawed protein or a time/temperature-incubated protein? The authors are required to furnish details on the buffer and incubation terms used in the methodology section of SEC-MALS.

Response: We carried out SEC-MALS experiment using a freshly thawed protein. We added the details on the buffer and incubation terms used in the methodology section of SEC-MALS. (p20, line 452-453)

Comment 4: During the C57D/C146D mutant purification steps, does the final gel filtration chromatogram show more than one peak or elution corresponding to a higher molecular weight? The authors requested to justify using the purification SEC chromatogram.

Response:

There was no peak corresponding to the higher molecular weight of the SOD1 protein in the final gel filtration chromatogram during the C57D/C146D mutant SOD1 purification steps, as seen in the figure below. Only the peak of the cleaved His-tag was found after the SOD1 peak.

UV chromatogram of size exclusion chromatography during C57D/C146D mutant SOD1

purification.

Comment 5: "...the monomeric feature of the mutant SOD1 is different from the disulfide-reduced wild-type SOD1." This sentence questions the use of the mutated C57D/C146D form of SOD1. Please justify.

Response: We realized that the sentence the reviewer commented was not fully explained. We intended to tell that the disulfide-reduced wild-type SOD1 was in the dimeric form, unlike the C57D/C146D mutant protein in our original manuscript. We tried to say the disulfide reduction of wild-type SOD1 is not the sole agent of SOD1 dimer destabilization. the destabilization of SOD1 dimer is induced by various events. For example, a previous research reported that the binding of metal ion prevents the dimer dissociation of disulfide-reduced SOD1¹.

In fact, we conducted the SEC-MALS in the presence of reducing agents to induce the disulfide-reduced SOD1 wild-type. The wild-type SOD1 was dimer even in the reducing condition, unlike the C57D/C146D mutant SOD1. Previous study from another group also explained that wild-type holo SOD1 is dimeric in disulfide reduced state². We added this data in **supplementary Fig. 2B** and updated the description in the revised manuscript. (p6, line 102-103)

Comment 6: CD spectra do not show too much deviation from the control. Does it justify the fibril formation due to the open conformation of the C57D/C146D mutant?

Response: According to the calculated contents of secondary structure based on the CD spectra, the large conformational changes were induced by the mutation. We added the calculated content of secondary structure along with CD spectrum. (Fig. 1D).

We admit that only the CD spectra cannot justify the fibril formation due to the open conformation of the C57D/C146D mutant. Thus, we added the SEC-MALS, thermal shift assay, and proteolytic sensitivity to support the conclusion. Most importantly, we presented the crystal structures of the mutant and wild-type SOD1 protein as the key results. Our crystal structures showed the large conformation shift in the loop IV by the developed charges due to the overoxidation at the cysteine residues.

Comment 7: Why was the thermal shift assay carried out in a different buffer system than the one it was stored in? If it has buffer-induced variations in thermal stability, are they not affected during the aggregation experiments or storage of the protein itself? Please justify.

Response: We were worried about the baseline destabilization from the Tris buffer. To address the reviewer's comment, we carried out the thermal shift assay using in the same buffer system used in the storage buffer: Tris buffer. We obtained the essentially same results in Tris buffer, as seen below.

We changed the graph to the thermal shift assay data using Tris buffer in this revision. (Fig. 1C)

Comment 8: "...structural features distinct from the reduced form of wild-type SOD1 with free thiols at Cys57 and Cys146." How do the authors confirm that the wild type has free thiols at C57 and C146 at 5 mM DTT? The previous results in Fig 1 suggest that the metal ions are intact, and there is no proteolysis at all due to the reduction of thiols at the above residues? Doesn't the reduction of an intra-disulfide bridge require higher DTT? Also, at 5 mM DTT, if the residues are at free thiols, it should behave similarly to the C57D/C146D mutant. Please justify.

Response: The comparison of C57D/C146D mutant SOD1 to the reduced form of wild-type SOD1 was intended to analyze the difference between the overoxidation of Cys57 and Cys146 and the disulfide cleavage without overoxidation.

DTT at a concentration of 5 mM is generally used for disulfide reduction of various proteins dissolved in solution, and previous studies have reported that 5 mM DTT is also effective in the disulfide reduction of SOD1³⁻⁵. In addition, using higher concentration of DTT above 5 mM was undesirable for the trypsin digestion as the protease activity of the trypsin may decrease in higher concentration of reducing agents.

To clearly compare the cysteine overoxidation with disulfide cleavage, C57A/C146A mutant SOD1 was created to compare proteolytic resistance with C57D/C146D mutant SOD1 or wild-type SOD1 protein. The C57A/C146A mutant SOD1 was digested less than C57D/C146D mutant SOD1 and more than wild-type SOD1 when incubated with trypsin for 30 min. This result demonstrates that the overoxidized SOD1 behave differently to the disulfide cleaved SOD1. (supplementary Fig. 2D, p6, line 123-124)

Comment 9: There is the usage of different concentrations of DTT throughout the work. Please justify the usage of the different concentrations of DTT in each of the experiments when used.

Response: We used 5 mM DTT for the trypsin digestion experiment because the trypsin activity is reduced above 5 mM. To make sure that the disulfide bond is reduced, we added the C57A/C146A mutant data in this revision (**Supplementary Fig.2D**). In the filament forming assay, we used 50 mM DTT. To keep the reducing power for several days, higher concentrations of DTT was required in the filament forming assay.

Comment 10: The spelling of ATN-224 in the text should be corrected.

Response: We corrected the spelling of ATN-224 in the text. (**p21, line 489 and p22, line 492**)

Comment 11: The statistical Significance value should be calculated and mentioned in Figure 1F.

Response: We added the statistical significance value in **Figure 1F**.

Comment 12: Figure 2A – Justify the presence of SOD1 monomers for pcDNA3.1 and SOD1 wild-type.

Comment 13: Figure 2A - The authors are required to provide quantification and significance, implying statistics in the Western blot figure.

Comment 14: “..the overoxidation mimicking C57D/C146D mutant SOD1 exhibits a higher propensity to form filaments in the cellular environment.” – The lane corresponding to this is not showing the formation of higher aggregates. Please justify.

Comment 15: Figure 2A – Based on the results shown in the previous Figure 1A, the WB blot should be exhibiting higher aggregates, irrespective of the presence or absence of the metal chelators. Please justify.

Comment 16: The results for Figure 2A are required to be explained in detail in the manuscript.

Comment 17: Previous studies specify that the loss of metals through chelation causes aggregation-prone effects. Please justify why the Western blotting results do not show any for wild-type?

Response to Comments 12-17: We cross-linked the cell lysate with glutaraldehyde to detect the SOD1 oligomers in the original WB analysis. We admit that Fig. 2A had many concerns in terms of quantification of the oligomeric SOD1 using the pan-SOD1 antibody. For the WB analysis, we had to denature the SOD1 protein during the SDS-PAGE with the cell lysate. Thus, we realized that we cannot distinguish the misfolded SOD1 protein with the pan-SOD1 antibody. To overcome this concern, we replaced the Western blotting with the dot blotting assay using the misfolded-SOD1-antibody (**Fig. 2A**). We believe that the dot blotting assay using the anti-misfolded antibody shows the only the misfolded SOD1 protein in the cell lysate

because the dot blotting assay did not contain the denaturing step. (Fig. 2A and p7, line 139-149)

Comment 18: The authors have used the pan-SOD1 antibody to assess the formation of aggregates and not a misfolded SOD1 antibody. The authors are required to reassess the WB results with the misfolded SOD1 antibody.

Response: We performed the dot blot assay using the misfolded SOD1 antibody. We replaced the Western blot image in **Figure 2A** with dot blot images using misfolded SOD1 antibody. (Fig. 2A and p7, line 139-149)

Comment 19: Figures 2A and 2B do not correlate for wild-type or C57D/C146D mutant in the presence of ATN-224. Please justify.

Response: It has been known that Zn²⁺ is bound to the SOD1 protein more strongly and is more important in the structural stability than Cu²⁺. According to the dot blotting assay results, the Zn²⁺ sequestering by TPEN promoted the filament formation of both wild-type and mutant protein, which showed the strong correlation to Fig. 2B. The Cu²⁺ sequestering by ATN-224 did not affect the filament formation compared to the chelator-free condition, which is largely correlated to Fig. 2B although the difference between the wild-type and the mutant seemed not significant. However, the variation at the Cu²⁺ sequestering between the different methods might not be important since this study is focused on the over-oxidation at the cysteine residues. Moreover, we consistently observed the Zn²⁺ chelation effect in both the dot blotting assay and the cellular inclusions in the cells.

Comment 20: Statistical quantification is required for the immunofluorescence images. The authors are requested to write in detail the same in the text as well.

Response: We added statistical quantification and detailed descriptions of immunofluorescence images to the manuscript. (**Fig. 2B and p7-8, line 150-161**)

Comment 21: Supplementary Figure 3 – Please mention which map (2Fo-Fc map or difference Fourier map or Omit map) is shown here (legend and text).

Response: We added an explanation that the 2Fo-Fc map was shown in supplementary figure 3 in the revised manuscript. (**p8, line 172, and figure legend of supplementary Fig. 4C, 4D**) In addition, we have updated the partial model (chain M) of the wild-type SOD1 structure to better match the electron density map (**supplementary Fig. 4D**).

Comment 22: Figure 3A – The hCCS is marked wrongly in the figure.

Response: The hCCS protein consists of three domains: N-terminal domain (domain I), the central domain (domain II), and C-terminal domain (domain III). Notably, the central domain

is highly similar to SOD1, which helps hCCS bind to SOD1. The reviewer might have confused this central domain of hCCS with the structure of SOD1. To clarify the structural notation, we modified the color of hCCS to blue (**Fig. 3A**)

Comment 23: Electron density (Omit map or Polder map) of the copper and zinc ions is required to be shown for the C57D/C146D structure.

Response: We added Omit map of the copper and zinc ions for the C57D/C146D structure in supplementary Fig. 5A.

Comment 24: Supplementary Fig 4A – The superposition of the two models should be done with one chain to see dimeric angle variations as well.

Response: We changed the superposition of the two models to be done with one chain (chain A) to see dimeric angle variations as well (**supplementary Fig. 5B**)

Comment 25: “ ... structure to the wild-type SOD1 with an RMSD of 0.710 Å between 152 C α atoms (Supplementary Fig. 4A) (residues 2-253).”

The rmsd value is too small for such a large structural difference in the loop regions (Figure 3B and Suppl Figure 4A). How was the superposition performed? Performed with respect to one chain (A or B) or both? Performed on the structural region only with respect to one chain (A or B) or both? Write in detail.

Response: We appreciate to the reviewer’s comments. The RMSD was calculated by Pymol with the default settings. Now we realized that the outliered regions were excluded during the structural alignment, which produced the smaller RMSD value. As the reviewer suggested, we re-calculated the RMSD values on the entire SOD1 protomer (chain A) in the setting cycle value of 0 in Pymol. As a result, the RMSD value was 1.311, which appeared to represent the large structural difference of the loop regions. We revised the manuscript accordingly. (**p9, line 181-183**)

Comment 26: “On the structural superposition with the wild-type dimeric structure...” the superposition indicates loss of G51 interaction but with a missing loop density. Figure 3A shows no missing loop in the second chain. The inference is invalid as the succeeding missing loop makes it ambiguous.

Response: We acknowledge the concerns of the reviewer. To address the reviewer’s comment, we deleted Fig. 3D and its associated description from the revised manuscript.

Instead, we added an explanation that the open conformation of loop IV and the detachment of the loop IV from the beta barrel of SOD1 would destabilize the SOD1 dimer, since the loop IV is in the dimer interface. According to the previous paper⁶⁻⁸, the closed conformation of loop IV is important for dimerization. Our findings clearly demonstrated that open conformation is induced by the double mutant C57D/C146D (**Fig. 3B and the newly added Fig. 3C**) Therefore, our results well support that the destabilization of dimer SOD1 is promoted due to the mutation or overoxidation in cysteine residues, even despite of the omitted Fig. 3D.

Comment 27: Supplementary Figure 4C – The side chain electron density of R144 is absent. Again, which electron density have shown here? The conformations of the sidechain of R144 in both the chains cannot be accounted for as there is no density in these regions. In the PDB file, the occupancy corresponding to these atoms was set to ‘zero’? Conclusions cannot be inferred from the missing density to evaluate disengagement of the residue, which is proposed to lead to superoxide catalytic activity deactivation. A biochemical assay on the R144 mutant is required to validate the inference.

Response: We intended to show the flexibility of the Arg143 side chain in C57D/C146D mutant SOD1 through no electron density map around the side chain in Fig. S4D, at the same contour level as in Fig. S4C. Thus, we displayed the 2FoFc map, contoured at 1.5 sigma, around the entire Arg143 residue. We refined the temperature factors after the occupancy set to 1 to represent the flexibility of each atom.

To support the flexibility of R143 in C57D/C146D mutant SOD1, we compared the Wilson B-factor of wild-type and C57D/C146D mutant SOD1 structure. The B-factor of Arg143 side chain ($\sim 41 \text{ \AA}^2$) was similar to the overall B-factor (42 \AA^2) of the entire wild-type SOD1 coordinates, whereas the B-factor of Arg143 side chain ($\sim 50 \text{ \AA}^2$) was higher than the overall B-factor (17 \AA^2) of the entire C57D/C146D SOD1 coordinates.

According to the previous study⁹, a biochemical assay was performed to show that Arg144 is important for the catalytic activity of SOD1. So additional biochemical assay on the Arg144 mutant is not required.

In addition, we corrected Arg144 into Arg143 in **Supplementary Fig. 4C**.

Comment 28: It is advised to the authors to maintain the same chain colors for Supp. Fig 4 and Fig. 3 maintain a clear understanding of the structures.

Response: We changed **Supplementary Fig. 4** and **Fig. 3** to maintain the same chain colors to maintain a clear understanding of the structures.

Comment 29: Data analysis of ThT measurements should be included in the methodology.

Response: We added descriptions for data analysis of ThT measurements in the methodology. (p23, line 530-532)

Comment 30: Was the ThT setup performed for three individual experimental replicates or triplicate within the same plate?

Response: We performed all the ThT experiments at least ten sets in each experiment of the slight variations in the experiment conditions were included, even though the replicate numbers are different among the experiments.

Comment 31: Why is there a difference in the ThT aggregation times without and with DTT for the C57D/C146D over-oxidation mutant?? Justify.

Response: In addition to Cys57 and Cys146 forming the intramolecular disulfide, there are two additional cysteine residues (Cys6 and Cys111). The presence of DTT may have kept Cys6 and Cys111 in a reduced state during the filament formation. In contrast, without DTT, Cys6 and Cys111 could be oxidized during the filament formation assay. We think that DTT affected the filament formation of the C57D/C146D over-oxidation mutant through the Cys6 and Cys111.

Comment 32: Why were a different pre-incubation and buffer conditions used for the C6/C111 mutant as compared to the major C57D/C146D mutant? The ThT graphs for the C6/C111 mutant in comparison with the wild-type should also be shown. If both the mutants are meant to cleave intra-disulfide bonds, the condition of amyloid formation assessment should not vary. The authors need to reassess the experimental conditions used for the ThT analysis with C6-C111 mutant.

Response: We used the same pre-incubation and buffer conditions for the C6/C111 mutant as compared to the major C57D/C146D mutant (see 'Filament formation of the SOD1 proteins' in the Methods section and Fig. 4B). We have shown the ThT graphs for the C6/C111 mutant in comparison with the wild-type SOD1 (see Fig. 4B in the original manuscript). We believe that the cleavage at the intramolecular disulfide bond in loop IV is important in the filament formation, but not the (possibly intermolecular) disulfide bond involved in Cys6 and Cys111. The C6A/C111A showed the higher filament formation than the wild-type SOD1 protein, which indicated that the disulfide bond involved in Cys6 and Cys111 interfered with the filament formation in contrast to the Cys57 and Cys146.

References

- 1 Furukawa, Y. & O'Halloran, T. V. Amyotrophic lateral sclerosis mutations have the greatest destabilizing effect on the apo-and reduced form of SOD1, leading to unfolding and oxidative aggregation. *Journal of Biological Chemistry* **280**, 17266-17274 (2005).
- 2 Lindberg, M. J., Normark, J., Holmgren, A. & Oliveberg, M. Folding of human superoxide dismutase: disulfide reduction prevents dimerization and produces marginally stable monomers. *Proceedings of the National Academy of Sciences* **101**, 15893-15898 (2004).
- 3 Chattopadhyay, M. *et al.* Initiation and elongation in fibrillation of ALS-linked superoxide dismutase. *Proceedings of the National Academy of Sciences* **105**, 18663-18668 (2008).
- 4 Chan, P. K. *et al.* Structural similarity of wild-type and ALS-mutant superoxide dismutase-1 fibrils using limited proteolysis and atomic force microscopy. *Proceedings of the National Academy of Sciences* **110**, 10934-10939 (2013).
- 5 Chattopadhyay, M. *et al.* The Disulfide Bond, but Not Zinc or Dimerization, Controls Initiation and Seeded Growth in Amyotrophic Lateral Sclerosis-linked Cu,Zn Superoxide Dismutase (SOD1) Fibrillation. *J Biol Chem* **290**, 30624-30636, doi:10.1074/jbc.M115.666503 (2015).
- 6 Danielsson, J., Kurnik, M., Lang, L. & Oliveberg, M. Cutting off functional loops from homodimeric enzyme superoxide dismutase 1 (SOD1) leaves monomeric β -barrels. *Journal of Biological Chemistry* **286**, 33070-33083 (2011).
- 7 Banci, L., Bertini, I., Cramaro, F., Del Conte, R. & Viezzoli, M. S. The solution structure of reduced dimeric copper zinc superoxide dismutase: the structural effects of dimerization. *European Journal of Biochemistry* **269**, 1905-1915 (2002).
- 8 Wright, G. S., Antonyuk, S. V. & Hasnain, S. S. The biophysics of superoxide dismutase-1 and amyotrophic lateral sclerosis. *Quarterly reviews of biophysics* **52** (2019).
- 9 Borders Jr, C. L., Saunders, J. E., Blech, D. & Fridovich, I. Essentiality of the active-site arginine residue for the normal catalytic activity of Cu, Zn superoxide dismutase. *Biochemical Journal* **230**, 771-776 (1985).

REVIEWERS' COMMENTS:

Reviewer #1 (Remarks to the Author):

The authors have revised except for one important change requested. In concluding section, the change that authors have made is not sufficient. The sentence "Our findings would explain a significant number of sportsmen with sALS" should be replaced by "Oxidative stress that is generally increased in sports that demand extreme physical exercises may contribute to recent reports of a significant number of sportsmen with sALS, where over-oxidation of disulfide residues may trigger the pathogenicity".

Reviewer #2 (Remarks to the Author):

My suggestions and comments were appropriately improved. Comparison with C57A/C146A mutant, originally prepared, would be helpful for discussion in more detail. I have no further comment for accepting this manuscript.

Reviewer #3 (Remarks to the Author):

In response to the queries, the authors have addressed them appropriately in the revised manuscript. Hence, the manuscript can be considered for publication.

Point-by-point response to reviewers' comments

Reviewer #1

Comment: The authors have revised except for one important change requested. In concluding section, the change that authors have made is not sufficient. The sentence "Our findings would explain a significant number of sportsmen with sALS" should be replaced by "Oxidative stress that is generally increased in sports that demand extreme physical exercises may contribute to recent reports of a significant number of sportsmen with sALS, where over-oxidation of disulfide residues may trigger the pathogenicity."

Response: As the reviewer suggested, we replaced the sentence in concluding section "Our findings would explain a significant number of sportsmen with sALS" by "Oxidative stress that is generally increased in sports that demand extreme physical exercises may contribute to recent reports of a significant number of sportsmen with sALS, where over-oxidation of disulfide residues may trigger the pathogenicity."

Reviewer #2

Comment: My suggestions and comments were appropriately improved. Comparison with C57A/C146A mutant, originally prepared, would be helpful for discussion in more detail. I have no further comment for accepting this manuscript.

Response: We appreciate for the reviewer's comments.

Reviewer #3

Comment: In response to the queries, the authors have addressed them appropriately in the revised manuscript. Hence, the manuscript can be considered for publication.

Response: We appreciate for the reviewer's comments.